# Astrocytes modulate neuronal development by S100A6 signaling

Valentina Cinquina [1], Evgenii O. Tretiakov [1], Predrag Kalaba[2], Alán Alpár[3,4], Daniela Calvigioni[1,11], Fabiana Piscitelli[5], Erik Keimpema [1], Vincenzo Di Marzo[5,6,7], Alexej Verkhratsky[8,9] & Tibor Harkany [1,10] ✉

Neuronal morphogenesis relies on intercellular signaling. Astrocytes release metabolites, trophic, and guidance factors to promote neuronal maturation. In contrast, the mechanisms by which astrocytes could limit and stabilize neuronal connectivity remain less explored. Here, we find cortical astrocytes to express and release S100A6, a $Ca^{2+}$-binding protein ('calcyclin'). Simultaneously, the majority of cortical neurons expressed calcyclin-binding protein (CaCyBp), a bona fide binding partner for S100A6. In neurons, CaCyBp maintains the unfolded protein response pathway, thereby controlling proteostasis. When released, S100A6 inhibits CaCyBp-mediated signaling, thus slowing protein turnover, and, consequently, neuritogenesis. In the cerebral cortex of male mice, S100A6-CaCyBp signaling during gestation is sensitive to the mother's nutritional status, particularly eicosapentaenoic acid intake. Thus, a member of the S100 protein family acts as an astroglia-derived morphogen, whose action on neurons is modulated by environmental factors.

Reciprocal communication with astrocytes underpins the development, survival, and excitability of neurons. Therein, astrocytes release gliotransmitters (such as canonical neurotransmitters, adenosine/ATP, and trophic factors), neurotransmitter precursors (glutamine, L-serine), and metabolites (chiefly lactate and pyruvate), and also clear excess neurotransmitters and waste products, including potential neurotoxins[1]. During brain development, astrogliogenesis coincides with synaptogenesis[1]. Astrocytes are broadly known to express and release differentiation and survival factors (e.g., brain-derived neurotrophic factor, glial cell line-derived neurotrophic factor, thrombospondin, hevin/Sparcl1) to promote the polarization, morphological differentiation, and synaptogenesis of neuroblasts[1]. In contrast, astrocyte-derived factors that are inhibitory to neuritogenesis, thus acting to balance growth *vs*. synapse selection and stabilization, remain less explored.

Considering that major steps of cellular differentiation are regulated by $Ca^{2+}$ signals (be this the activation of kinases, cytoskeletal instability, directional growth decisions, synaptogenesis)[1], $Ca^{2+}$-binding and sensor proteins appear particularly important, considering that some can even undergo regulated release to support intercellular interactions[2]. In astrocytes, S100β, a $Ca^{2+}$-binding protein within the EF-hand superfamily, is best known as one of the 25 members of the

[1]Department of Molecular Neurosciences, Center for Brain Research, Medical University of Vienna, Vienna, Austria. [2]Institute of Biological Chemistry, Faculty of Chemistry, University of Vienna, Vienna, Austria. [3]SE NAP Research Group of Experimental Neuroanatomy, Semmelweis University, Budapest, Hungary. [4]Developmental Biology, Department of Anatomy, Histology, and Embryology, Semmelweis University, Budapest, Hungary. [5]Endocannabinoid Research Group, Institute of Biomolecular Chemistry (ICB), National Research Council (CNR), Pozzuoli, Italy. [6]Canada Excellence Research Chair on the Microbiome-Endocannabinoidome Axis in Metabolic Health, Institut Universitaire de Cardiologie et de Pneumologie de Québec, Université Laval, Québec, QC, Canada. [7]Institut sur la Nutrition et les Aliments Fonctionnels, Centre Nutriss, Université Laval, Québec, QC, Canada. [8]Faculty of Biology, Medicine and Health, The University of Manchester, Manchester, UK. [9]Achucarro Centre for Neuroscience, IKERBASQUE, Basque Foundation for Science, Bilbao, Spain. [10]Department of Neuroscience, Biomedicum 7D, Karolinska Institutet, Solna, Sweden. [11]Present address: Department of Neuroscience, Biomedicum 4B, Karolinska Institutet, Solna, Sweden. ✉e-mail: Tibor.Harkany@meduniwien.ac.at

S100 protein family[2] because it is constitutively expressed in most astrocytes across brain regions[3,4], and associated with inflammation in adults[3,5]. However, if S100 proteins are expressed in astrocytes during brain development and could be assigned to regulate neuronal morphogenesis remains unknown.

Here, we explored the developmental expression patterns of S100 family proteins by processing open-label single-cell RNA-seq data during mouse fetal brain development, and identified S100A6/calcyclin, its binding partner (calcyclin-binding protein; CaCyBp), and the main downstream targets *Siah1/Siah2*, E3 ubiquitin ligases, with precisely timed expression coinciding with neuritogenesis and synaptogenesis in the late embryonic/neonatal period in mice. We then used in situ hybridization and immunohistochemistry to map S100A6 and CaCyBp to astrocytes and neurons, respectively, identified S100A6 as a secreted ligand by mass spectrometry, protein biochemistry, and neurochemistry in cellular models in vitro, and combined genetic and pharmacological tools to demonstrate that S100A6 could inhibit neuritogenesis, likely by slowing protein turnover. Notably, we found both S100A6 and CaCyBp expression to be sensitive in the developing fetal nervous system to nutritional imbalance, particularly to eicosatetraenoic acid (EPA) availability. Subsequently, we have associated the deregulation of S100A6-CaCyBp signaling between astrocytes and neurons with maternal exposure to hypercaloric/ω3-polyunsaturated fatty acid (PUFA)-enriched diets in male mice. Cumulatively, these data suggest that S100A6, once released, affects neuronal morphogenesis, at least in the mouse brain.

## Results

### S100 proteins in the developing mouse brain

First, we addressed the expression sites and developmental dynamics of S100 proteins in the mouse brain by processing open-label single-cell RNA-seq data on the fetal somatosensory cortex[6]. During corticogenesis (that is, from embryonic day (E)10 to postnatal day (P)4), cell identities were mapped in age-matched tissues by UMAP representation (Fig. 1a–d). Progenitors, as well as astrocytes, were subdivided into 16 clusters based on marker gene expression (see "Methods"; Fig. 1d). Fifteen of the S100 proteins were detected from E10.5 on (Fig. 1e). Four of the proteins (S100A1, S100A6, S100A13, and S100A16) gradually increased in expression in astrocytes (up to P4). Among these, *S100a6* (also termed "calcyclin") showed a sharp increase between E18.5 and P4, when astrogliogenesis peaks[1,6] (Fig. 1e). Detailed examination of the gene sets conferring cell identity in the single-cell RNA-seq data showed that S100A6 was chiefly expressed by astrocytes in neonatal and postnatal mouse cortices (Fig. 1f, g and Supplementary Fig. 1a). As such, UMAP representation revealed that *S100a6* expression matched with astrocytes also enriched in e.g., *S100b*[7], *Apoe*[8], *and Aldh1l1*[9,10] (Supplementary Fig. 1b). qPCR and Western blotting validated the presence of *S100a6* mRNA ($n = 4$–7 male mice/time point, $p = 0.003$ for E14.5 *vs.* P4, one-way ANOVA followed by Tukey's multiple comparisons test) and protein expression ($n = 3$ male mice/time point; $p = 0.017$ for E18.5 *vs.* P4, Student's *t*-test), respectively, during development (Supplementary Fig. 1c, d). These data suggest that S100A6 could control a specific event during brain development, noting that its expression can be maintained in mature astrocytes postnatally[11].

### S100A6 and calcyclin-binding protein localization

S100A6 promotes cell proliferation and differentiation, and protects from noxious insults[12,13]. S100A6 contains two EF-hand $Ca^{2+}$-binding motifs, which are helix-loop-helix structural domains (Supplementary Fig. 1e). Upon $Ca^{2+}$ binding, S100A6 undergoes a conformational change that exposes a hydrophobic region to allow for its interaction with target proteins. S100A6 binds to the C-terminus of calcyclin-binding protein (*Cacybp* or *S100a6bp*) to promote its interaction with *Siah1a/Siah1b*, E3 ubiquitin ligases, through its N-terminal domain

(Supplementary Fig. 1f). Thus, S100A6, if released from, e.g., astrocytes, could regulate protein degradation/turnover in neighboring cells, including neurons, alike in other tissues[14,15], and malignant cells[7].

Single-cell RNA-seq placed *Cacybp* in both apical progenitors and postmitotic neurons (Fig. 1e, h, i and Supplementary Fig. 1g). When testing the availability of *Cacybp* in neurons, we found it ubiquitously expressed at E10.5, with stable levels from E14.5 to P4 (Fig. 1e, h and Supplementary Fig. 1g). *Siah1a* and *Siah1b* were also detected by single-cell RNA-seq (Fig. 1e). At E18.5, when *S100a6* expression begins to increase in the astrocyte lineage (Fig. 1e, g), UMAP assigned *Cacybp* expression to virtually all neuronal lineages in the mouse cortex (note its limited expression in astrocytes for cell-autonomous function; Fig. 1i). CaCyBp expression at both the mRNA and protein levels was confirmed by qPCR (Supplementary Fig. 1h; $n = 5$–7 male mice/time point, $p = 0.009$ for E14.5 *vs.* P4, one-way ANOVA followed by Tukey's multiple comparisons test) and Western blotting (Supplementary Fig. 1i; $n = 3$ male mice/time point, $p = 0.040$ for E18.5 *vs.* P4, Student's *t*-test). Both methods demonstrated stable levels of *Cacybp* mRNA and CaCyBp protein during development, followed by a significant increase postnatally.

### Complementarity of S100A6 and calcyclin-binding protein expression

To map S100A6 and CaCyBp in brain tissue, we performed in situ hybridization experiments on coronal sections of E18.5 and P4 mouse brains ($n = 3$ male mice/time point) using probes specific for these markers (*see* "Methods"). In the somatosensory cortex (ssCtx; Fig. 2a), co-localization between *S100a6* and *S100b*[7], *Apoe*[8], *Aldh1l1*[9,10], and *Gfap* mRNAs, pleiotropic markers of astrocytes, was found in the cortical plate at both E18.5 and P4 (Fig. 2a, d and Supplementary Fig. 2).

During the same developmental window, *Cacybp* expression was mostly confined to cells that co-expressed the neuronal marker *Rbfox3* (which encodes NeuN; Fig. 2b, e and Supplementary Fig. 3a, b; $n = 3$ male mice/time point throughout). The neuronal identity of *Cacybp*+ cells was further confirmed by immunohistochemistry, showing CaCyBp expression in both excitatory and inhibitory neurons in the ssCtx of mice at both E18.5 and P4 (Fig. 2c, f and Supplementary Fig. 3c, d). At E18.5, 96.3% of the cells that populated the ssCtx co-expressed both CaCyBp and microtubule-associated protein (MAP2), a neuronal marker. Moreover, 17.0% of neurons were positive for both CaCyBp and calretinin (CR, Fig. 2c *upper*; Supplementary Fig. 3c; $n = 3$ male mice/time point). Similar results were found at P4, when 90.8% of cells co-labelled for CaCyBp+/MAP2+ and 19.1% for CaCyBp+/CR+ (Fig. 2c *bottom*; Supplementary Fig. 3d; $n = 3$ male mice/time point). Likewise, CaCyBp was observed in the somatodendritic axis of neurons in the cortical plate of human fetuses during the second trimester (Fig. 3a; gestational week (GW) 27; male subject). Similar results were obtained in the adult cerebral cortex of both mice (Supplementary Fig. 3e; P120; male) and humans (Supplementary Fig. 3f; age: 60 years, male subject).

We then used primary cell cultures enriched in either neurons or astrocytes to support the cell-type-specific enrichment of both *S100a6* and *Cacybp* (Supplementary Fig. 3g). The *S100a6:Cacybp* mRNA (Fig. 3b) and protein (Fig. 3c) ratios were ~15:1 in astrocytes, whereas ~1:18 in neurons. Thus, we reveal the mutually exclusive expression of *S100a6* and *Cacybp* in astrocytes *vs.* neurons during corticogenesis, and suggest that intercellular S100A6-to-CaCyBp signaling could constitute a mode of communication if S100A6 is indeed a released substance.

### S100A6 is released from astrocytes

For S100A6 to be classified as a ligand released from astrocytes, among others cell types, its expression shall be regulated (e.g., by neurotransmitters and neuromodulators) and undergo activity-dependent release. We tested these properties in vitro, aided by

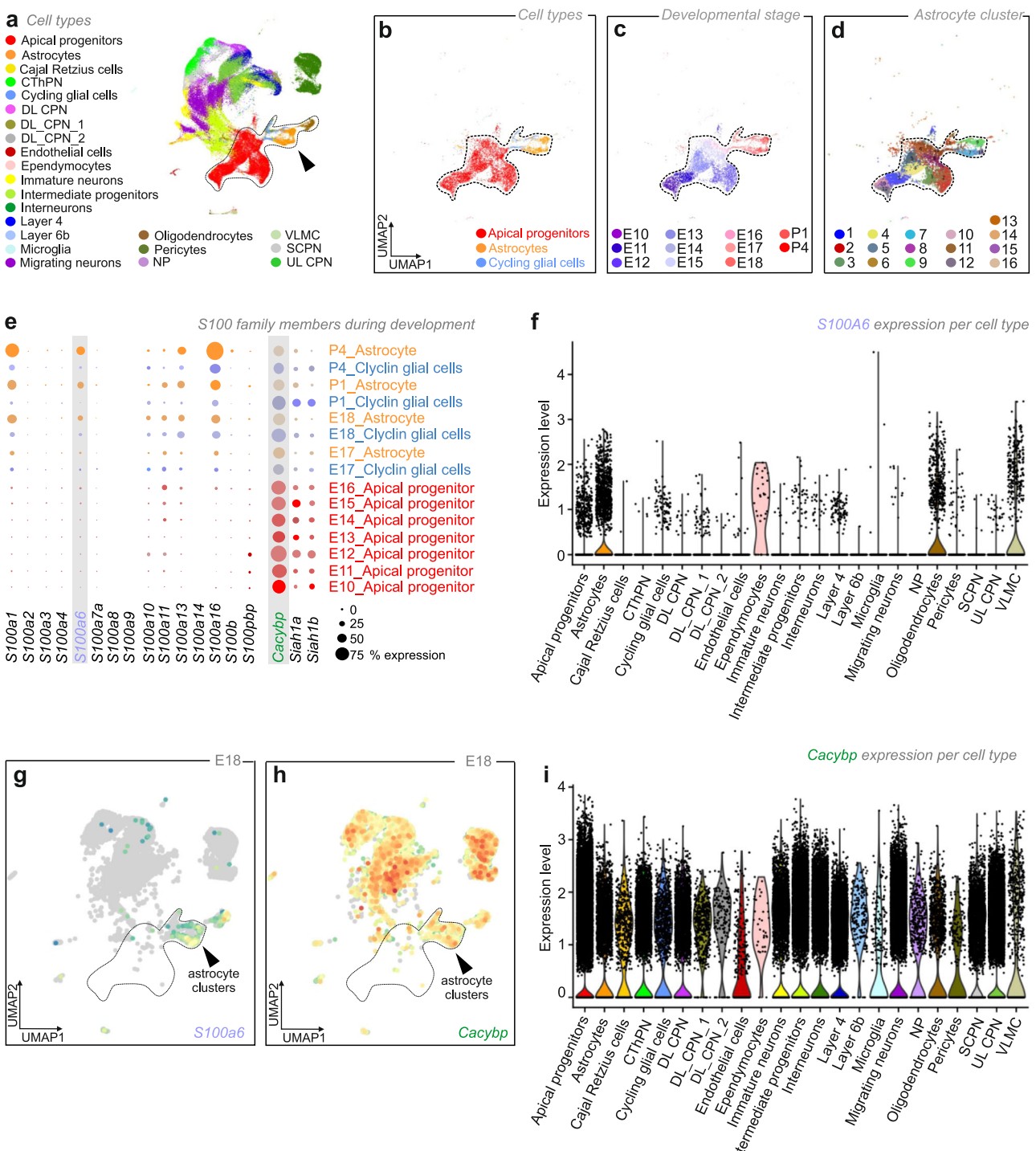

**Fig. 1 | S100A6 and CaCyBp in the developing mouse brain. a** UMAP visualization of single-cell RNA-seq data from the somatosensory cortex during mouse brain development (from embryonic day (E) 10 to postnatal day (P) 4), that were dis-aggregated by cell type. **b–d** Identification of cell types (**b**), developmental stages (**c**), and astrocyte clusters (**d**) during corticogenesis. **e** Single-cell RNA-seq-based developmental expression patterns for S100 proteins, calcyclin-binding protein (*Cacybp*), and *Siah*1a/b E3 ubiquitin ligases at the developmental time-points indicated. **f** Violin plot showing the expression of *S100a6* mRNA in cell types. **g** Cell type assignment for *S100a6* by UMAP at E18. **h** Cell type assignment for *Cacybp* by UMAP at E18. **i** Violin plot showing the expression of *Cacybp* per cell lineage.

IncuCyte-assisted live-cell imaging to control cell viability, by exposing primary astrocytes to glutamate (100 μM) for up to 24 h. Glutamate triggered S100A6 accumulation in the culture medium, with a ~fourfold increase by 24 h (Fig. 4a; *n* = 3 biological replicates; mixed for sex; *p* = 0.044 [0 h] *vs.* [6 h]; *p* < 0.0001 [0 h] *vs.* [24 h], *p* = 0.001 [6 h] *vs.* [24 h]; one-way ANOVA followed by Tukey's multiple comparisons). We propose this to be regulated S100A6 release

given the lack of cell death (Fig. 4b and Supplementary Fig. 4a; *see* "Methods" for the 3-(4,5-Dimethylthiazol-2-yl)-2,5-diphenylte-trazolium bromide (MTT) assay, and statistics), and the parallel (albeit disproportionate) increase in cytosolic S100A6 levels (within 6 h; Supplementary Fig. 4b; *p* = 0.021 [0 h] *vs.* [2 h]; *p* = 0.012 [0 h] *vs.* [6 h]; one-way ANOVA followed by Bonferroni's multiple comparison; *n* = 6 biological replicates, mixed for sex).

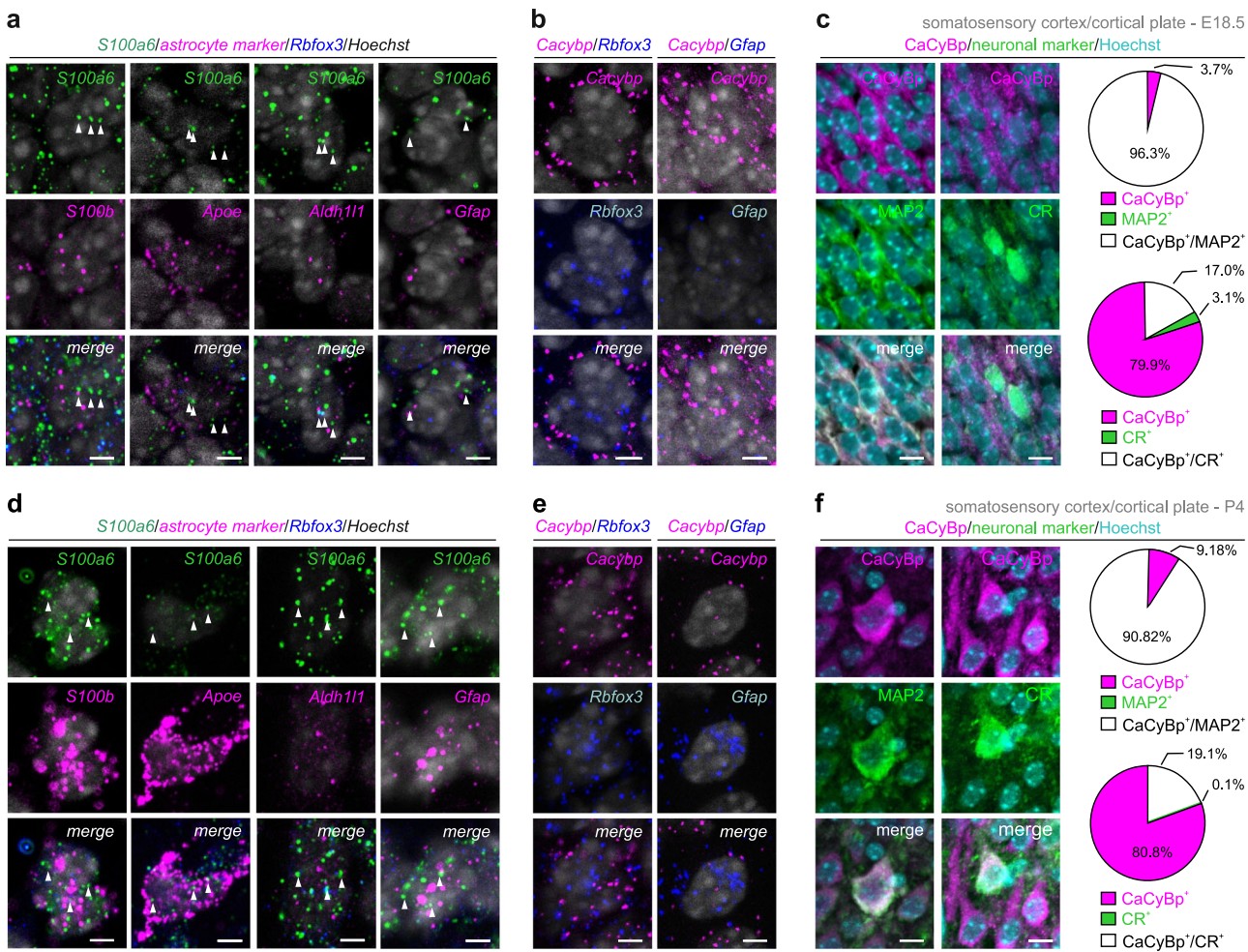

**Fig. 2 | Complementarity of S100A6 and calcyclin-binding protein expression in the cerebral cortex. a, d** *S100a6* mRNA expression overlapped with that of *S100b, Apoe, Aldh1l1*, and *Gfap* in both the fetal (E18.5; **a**) and neonatal (P4; **d**) mouse brain. Note the mutually exclusive expression of *S100a6* and *Rbfox3*, which encodes neuron-specific nuclear protein (NeuN). In contrast, *Cacybp* mRNA co-existed with that of *Rbfox3* but not *Gfap* at both time points in development (**b, e**).

**c, f** Representative images and quantification of CaCyBp localization by immuno-histochemistry in neurons of male fetal (E18.5; **c**) and neonatal (P4; **f**) mouse brains. Tissues were processed to simultaneously localize CaCyBp and either microtubule-associated protein 2 (MAP2; pan-neuronal marker) or calretinin (CR). Data were from *n* = 3 biological replicates. *Scale bars* = 12 μm.

Next, we asked if S100A6 could bind to CaCyBp by exposing Neuro-2a cells, which were invariably CaCyBp⁺ (Supplementary Fig. 4c), to His-tagged recombinant S100A6 (1 μg/μl; "pulse-chase experiment"). After 1 h, His-S100A6 was detected intracellularly in Neuro-2a cells by super-resolution microscopy (Fig. 4c). LC-MS/MS analysis supported these findings by detecting trypsin-cleaved peptide fragments specific to S100A6 in treated but not control Neuro-2a cultures (Fig. 4d and Supplementary Fig. 4d, e). Overall, these findings suggest a transcellular action analogous to those of trophic factors and morphogens[16], wherein astrocytes could serve as a potential source of S100A6 regulating neuronal responses by CaCyBp-based ligand recognition[17].

## S100A6 limits neuronal morphogenesis

We tested this hypothesis by exposing cortical neurons to recombinant S100A6 (1 μg/μl) for 4 days, which significantly reduced neurite complexity (Fig. 4e; *n* = 3 biological replicates; mixed for sex; *p* = 0.003; Student's *t*-test). This morphogenetic response coincided with reduced CaCyBp protein levels (Fig. 4f; *n* = 3 biological replicates; mixed for sex; *p* = 0.005; Student's *t*-test), which we qualified as a desensitization response. We then knocked down *Cacybp* expression in primary cortical neurons by AAV-mediated shRNA transduction. Attenuated *Cacybp* expression (at 15 days in vitro; Supplementary

Fig. 4f–h) reduced neurite outgrowth relative to mock-transfected controls (Fig. 4g; *n* = 23-27 cells; mixed for sex; *p* < 0.001 *vs*. mock-transfected controls; Student's *t*-test).

We then combined *Cacybp* knock-down with exposure to recombinant S100A6 (1 μg/μl) 24 h later. qPCR analysis showed an additive effect on both *Tubb3* (Fig. 4h; *n* = 5–8; mixed for sex; *p* = 0.045 [sh*Cacybp*] *vs*. [control]; *p* = 0.048 [His-S100A6] *vs*. [control]; *p* = 0.002 [sh*Cacybp* + his-S100A6] *vs*. [control]; one-way ANOVA followed by Tukey's multiple comparison; ANOVA summary: *F* = 6.757; *p* = 0.002) and *Actb* (Fig. 4h; *n* = 9–7; mixed for sex; *p* = 0.027 [sh*Cacybp*] *vs*. [control]; *p* = 0.008 [His-S100A6] *vs*. [control]; *p* = 0.005 [sh*Cacybp* + his-S100A6] *vs*. [control]; one-way ANOVA followed by Tukey's multiple comparison; ANOVA summary: *F* = 6.226; *p* = 0.002) following CaCyBp downregulation. Moreover, immunocytochemistry followed by single-cell reconstruction showed a reduction in neurite length (including the longest one designated as the putative "axon") after treatment (Fig. 4j; "axon": ANOVA summary: *F* = 11.88; *p* = 0.0004; *p* < 0.001 [sh*Cacybp*] *vs*. [control]; *p* = 0.003 [His-S100A6] *vs*. [control]; *p* = 0.002 [sh*Cacybp* + his-S100A6] *vs*. [control]; *neurites*: ANOVA summary: *F* = 7.16; *p* = 0.004; *p* = 0.010 [sh*Cacybp*] *vs*. [control]; *p* = 0.004 [His-S100A6] *vs*. [control]; *p* = 0.023 [sh*Cacybp* + his-S100A6] *vs*. [control]). The total number of neurites was not affected (Fig. 4i). These data

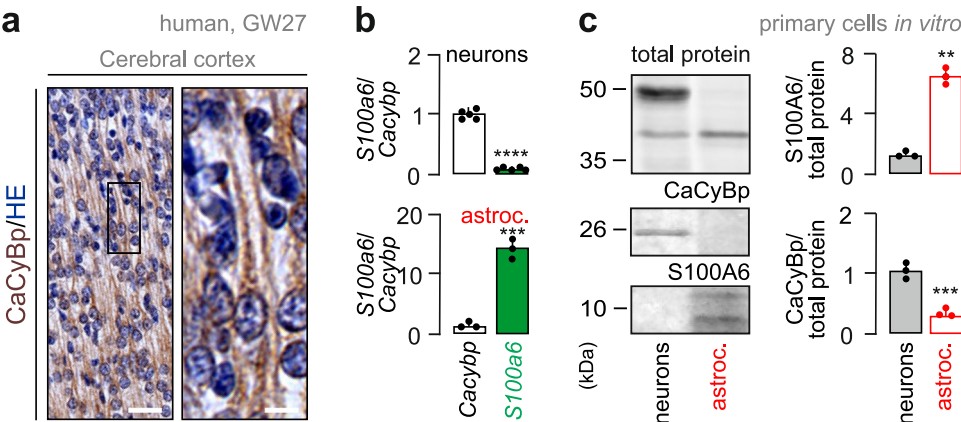

**Fig. 3 | CaCyBp in human fetal brain and its complementarity with S100A6 in vitro. a** CaCyBp immunoreactivity along the somatodendritic axis of cortical neurons in a human fetus at gestational week (GW) 27. The section was counter-stained with hematoxylin/eosin (HE; *scale bar* = 8 μm). **b**, **c** Comparison of *S100a6* and *Cacybp* mRNA (**b**) and protein (**c**) levels in cultured neurons *vs.* astrocytes. mRNA expression was normalized to *Gapdh*, a housekeeping gene. Two-tailed Student's *t*-test; means ± s.d. from *n* = 3 independent experiments (mRNA neurons: *n* = 6 biological replicates; ****p* < 0.0001. mRNA astrocytes: *n* = 3 biological repli-cates; ***p* < 0.001; protein S100A6: *n* = 3 biological replicates; ****p* < 0.0001; pro-tein CaCyBp: *n* = 3 biological replicates; ***p* < 0.001. Raw data are available in the Source Data File.

identify a role for S100A6 in limiting neuritogenesis through CaCyBp[14,18] (Fig. 4k).

Neuritogenesis relies on the rapid expansion of the cytoskeleton and its enrichment in specific proteins (receptors, transporters) along the neurite axis, including the prospective synapse. Subcellular frac-tionation placed CaCyBp in the cytosol, in particular, associated to the endoplasmic reticulum (ER; Fig. 5a). Electron microscopy confirmed this localization with CaCyBp often found in the proximity of the cis-ternae of the ER of neurons (Fig. 5b and Supplementary Fig. 5a). This location suggested that CaCyBp could be poised to affect proteostasis[19] (that is, protein synthesis and/or turnover at the ER).

## CaCyBp modulates protein turnover in neurons

Cortical neurons after *Cacybp* knock-down had reduced protein con-tent (*vs.* mock-transfected cells; Fig. 5c; *n* = 4 male mice; *p* = 0.005 *vs.* [mock-transfected control]; Student's *t*-test). Therefore, we tested the expression of molecular constituents of the ER stress pathway, whose activation increases protein degradation[20] by serving as effectors of the unfolded protein response (UPR) machinery[19]: G-protein coupled receptor 78 (*Grp78; p* = 0.001), cyclic AMP-dependent transcription factor ATF-6 alpha (*Atf6a; p* = 0.008), unspliced X-box-binding protein 1 (*Xbp1u; p* < 0.001) and spliced X-box-binding protein 1 mRNAs (*Xbp1s; p* = 0.002) were all downregulated (Fig. 5d; *n* = 6 biological replicates; mixed for sex; Student's *t*-test). This suggests the absence of a first phase (acute) cellular response to restore protein homeostasis. Besides cellular imaging (Fig. 4g−j), we used CCAAT-enhancer-binding protein homologous protein (CHOP)[21] as a surrogate to show that attenuated *Cacybp* expression did not trigger pro-apoptotic signaling (Fig. 5d; *p* = 0.701).

Subsequently, we tested the expression of ubiquitin specific peptidase 9, X-linked (USP9X), and ubiquitin specific peptidase 7 (USP7), both acting as deubiquitinating enzymes to prevent protein degradation by the proteasome[22], and thus being instrumental for cellular survival and function[23]. USP9X levels were significantly decreased (Fig. 5e**;** *p* = 0.008 *vs.* [mock-transfected control], *n* = 3 biological replicates; mixed for sex; Student's *t*-test). USP7 showed lesser reduction (Fig. 5e; *p* = 0.113 *vs.* [mock-transfected control]; *n* = 3 biological replicates). These results were compatible with increased tubulin acetylation (Fig. 5f; *p* < 0.001 *vs.* [mock-transfected control]; *n* = 3 biological replicates; mixed for sex; Student's *t*-test), a post-translational mark of long-lived microtubules, suggesting slowed cytoskeletal dynamics[24,25]. These changes were phenocopied by

sustained exposure to recombinant S100A6, which similarly reduced *Cacybp* expression (Supplementary Fig. 5b−f).

Subsequently, we asked if treating neurons with another S100 family member, S100A4, which was also expressed by astrocytes during brain development, even if more sporadically (Fig. 1e and Supplementary Fig. 6a), could mimic the morphogenetic effects of S100A6. It is noteworthy that a binding partner for S100A4 is not known, and CaCyBp does not bind S100A4[26]. S100A4 exposure did not affect neurite development in vitro (Supplementary Fig. 6b). Similarly, the expression of neither *Cacybp* nor of the molecular constituents of the ER stress pathway were altered (Supplemen-tary Fig. 6c).

Lastly, we asked if S100A6 localization within Neuro-2a cells could be affected by shRNA-mediated *Cacybp* knock-down. To this end, Neuro-2a cells were transfected with siRNA targeting *CaCyBp* for 96 h, with subsequent exposure to His-tagged S100A6 for an additional 1 h. Super-resolution microscopy (Fig. 5g) revealed S100A6 intracellularly. This is not unexpected because CaCyBp does not represent a cell-surface binding site that would limit S100A6 internalization. Yet we found reduced S100A6 localization within the ER fraction where CaCyBp was shown to accumulate (Fig. 5h). Thus, we outline an intracellular signaling cascade centered on the UPS machinery for S100A6 to limit neuronal morphogenesis (*see also* Fig. 4k).

## S100A6 signaling is sensitive to eicosapentaenoic acid

Next, we sought to address if environmental factors, particularly components of a maternal diet that is societally seen as "healthy" could affect S100A6-CaCyBp signaling in the developing brain. We hypo-thesized that eicosapentaenoic acid (EPA), an ω-3 polyunsaturated fatty acid (PUFA) that is particularly enriched in experimental diets (Supplementary Table 1), could be a candidate to affect S100A6 signaling because it modulates both neurogenesis[27,28] and astrocyte differentiation[29] by being incorporated in the membranes of glial cells[29–31]. However, a biological substrate for EPA is as yet unknown.

First, we tested the dose-dependence of EPA action on both neurons and astrocytes. By using high-throughput live-cell imaging of neurons for 7 DIV, we found that 1 μM but not 5 μM concentrations of EPA induced neuritogenesis in cortical neurons (including branching; Fig. 6a; *n* = 7 biological replicates; mixed for sex; *neurite length*: *p* < 0.0001 for both [control] *vs.* [EPA, 5 μM] and [EPA, 1 μM] *vs.* [EPA, 5 μM]; *neurite branches*: *p* < 0.0001 for both [control] *vs.* [EPA, 5 μM]

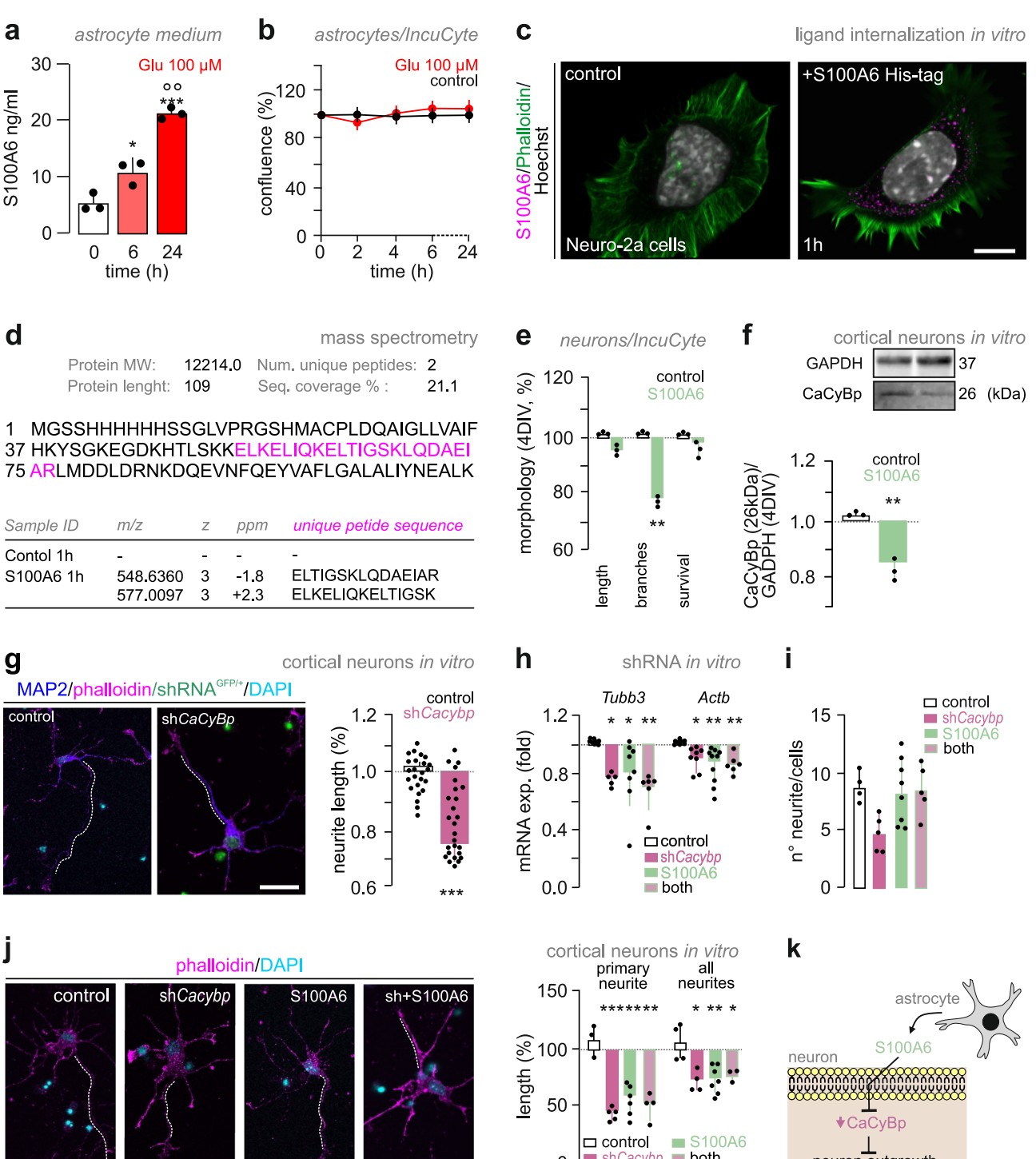

and [EPA, 1 µM] *vs.* [EPA, 5 µM]; Supplementary Fig. 7a). EPA did not impact astrocyte proliferation even at an excessive dose of 30 µM after 24 h (Fig. 6b and Supplementary Fig. 7b). No effect was seen on astrocyte viability either (Supplementary Fig. 7c). Yet, EPA increased the metabolic activity of astrocytes in the MTT assay (Supplementary Fig. 7c), which we interpreted as altered mitochondrial function likely upstream to ER stress[32]. Then we asked if EPA could be one of the effectors to modulate S100A6 secretion from astrocytes, particularly since increased cellular metabolism could underpin release events. Thus, we exposed astrocytes to EPA (30 µM) for 24 h and we found that it induced S100A6 release in vitro (Fig. 6c; $n = 3$ biological replicates; mixed for sex; $p < 0.0001$ [0 h] *vs.* [24 h], $p = 0.0004$ [6 h] *vs.* [24 h]). Both 5 µM and 10 µM EPA concentrations were also effective

(Supplementary Fig. 7d). Thus, we characterized S100A6 as secreted ligand possibly released upon EPA exposure, at least in vitro.

Next, we performed experiments in which conditioned media of astrocytes, collected 6 h after EPA exposure (30 µM), were transferred onto neurons (Supplementary Fig. 7e). After an incubation period of 24 h, neurons treated with conditioned media had reduced *Cacybp* mRNA levels (Fig. 6d; $n = 3$ biological replicates; mixed for sex; $p = 0.021$; Student's *t*-test), as well as a lowered expression of the molecular constituents of the ER stress pathway (Fig. 6e; $n = 3$ biological replicates; mixed for sex) in vitro.

Thereafter, we sequentially knocked down S100A6 expression in astrocytes by a pool of siRNAs (1 µM) for 96 h, exposed them to EPA (30 µM) for 6 h, and then collected conditioned media for transfer

**Fig. 4 | Astrocyte-derived S100A6 limits neuritogenesis. a** S100A6 ELISA in culture media upon glutamate-treated cells (100 μM) for 24 h ($n = 3$ biological replicates). **b** Astrocyte survival upon glutamate exposure. Data were normalized to cell density prior to treatment on day 1 in vitro (DIV, $n = 8$ biological replicates). **c** Neuro-2a cells treated with His-S100A6 (1 μg/μl) for 1 h and double-labelled for F-actin (phalloidin; green), anti-S100A6 (magenta), and DAPI (nuclear counterstain appears in white; $n = 3$ biological replicates). **d** Mass spectrometry confirmed S100A6 internalization in Neuro-2a cells after His-S100A6 treatment (21.1% sequence coverage; 2 unique peptides). S100A6 was absent in controls. The mass-to-charge ratio (m/z), charge state (z), error in parts per million (ppm), unique peptide sequences, and sequence coverage were defined. **e** In primary cortical neurons, S100A6 (1 μg/μl) altered neurite outgrowth over 4 days in vitro (DIV). Neurite length was normalized to baseline at 1 DIV. Cell survival refers to the surface occupancy of neurons ($n = 3$ biological replicates). **f** CaCyBp reduction in cortical

neurons after His-S100A6 treatment (1 μg/μl; 4 DIV) normalized to GAPDH ($n = 3$ biological replicates). **g** shRNA knockdown of CaCyBp reduced neurite length in cortical neurons at 15 DIV. Neurite length (μm) was normalized to mock-transfected controls ($n = 23$ [control cells]; $n = 27$ [sh*Cacybp* cells]). **h** *Tubb3* and *Actb* mRNA levels. Data were normalized to *Gapdh*. **i** Neurite numbers after shRNA and recombinant S100A6 exposures. **j** Reduced neuritogenesis 24 h after dual manipulation. Primary neurite and total neurite length (μm) were normalized to mock-transfected controls and expressed as a percentage. $n = 4$ [control; sh*Cacybp*]; $n = 7$ [His-S100A6]; $n = 3$ [sh*Cacybp*+His-S100A6]. **k** Schema of the proposed molecular pathway. All experiments included samples mixed for sex. *Scale bars* = 12 μm (**g, j**), 8 μm (**c**). Data were expressed as means ± s.d. throughout. Data were statistically evaluated by two-tailed Student's *t*-test (**p < 0.01, ***p < 0.001; **e, f, g**) or one-way ANOVA (*p < 0.05, **p < 0.001, ***p < 0.0001, *post-hoc* comparisons **a, h, i, j**). Raw data, including exact statistical parameters, are available in the Source Data File.

---

onto neurons (for 24 h; **7a**) to specify astrocyte-derived S100A6 as a mediator of EPA-induced growth effects on neurons. In astrocytes, both Western blotting (cytosol) and ELISA (medium) revealed attenuated S100A6 expression upon EPA treatment relative to either mock-transfected controls or EPA alone (Fig. 7b–d). In neurons, standard culture medium and astrocyte-conditioned medium (conditioning for 6 h only without treatment to the astrocytes; Supplementary Fig. 8a) led to a statistically significant increase in neuronal growth (Supplementary Fig. 8b). This effect is compatible with neuronal differentiation in response to growth factors released by astrocytes. From this point on, the morphological variables determined in neurons exposed to conditioned astrocyte medium served as baseline ("control"). In addition, we used direct stimulation by EPA (30 μM; Fig. 7a) to limit bias through its direct effect as "left-over contaminant" in conditioned media after 6 h. Once depleting conditioned media in S100A6 by RNAi, neurons underwent neuritogenesis (no mock-transfected controls at this point; Fig. 7e, f; $p = 0.042$ [control] *vs.* [sRNA alone]). Moreover, a differential, and statistically significant, neuritogenic response was observed when comparing EPA alone that reduced the length of the lead neurite *vs.* siRNA-mediated *S100a6* knock-down in conjunction with EPA stimulation ($p = 0.0006$ [conditioned with EPA] *vs.* [siRNA + conditioned with EPA]). Likewise, the cumulative length of neurites and their branching were rescued when knocking down *S100a6* before conditioning astrocytes with EPA (Fig. 7g; *neurite length*: $p = 0.0005$ [conditioned with EPA] *vs.* [siRNA + conditioned with EPA]; *branching*: $p = 0.0001$ [control] *vs.* [sRNA alone], and $p = 0.025$ [conditioned with EPA] *vs.* [siRNA + conditioned with EPA]). As before, increased S100A6 levels in the conditioned medium were coupled with a reduction of *Cacybp* mRNA expression in neurons (Supplementary Fig. 8c). Thus, we suggest that modulating S100A6 release impinges upon neuronal differentiation through CaCyBp engagement.

## Transgenerational modulation of S100A6 signaling

EPA is a key component of "ω-3 polyunsaturated fatty acid (PUFA)-enriched" diets in humans, which are preferred during pregnancy[33,34]. Therefore, we exposed female dams for 8 weeks to a hypercaloric diet enriched in ω-3 PUFAs prior to conception and throughout pregnancy (Supplementary Fig. 9a and Supplementary Table 1)[35] (EPA enrichment: ~fivefold *vs.* control diet).

We first determined cytoarchitectonic features of fetal cortices at E18.5 male fetuses after gestational exposure to the experimental diet. Wild-type C57Bl6/J dams were pulsed with 5-ethynyl-2′-deoxyuridine (EdU) on E14.5, the peak of cortically-oriented neurogenesis[36], with the distribution of EdU+ progeny in fetal cortices quantified and plotted on E18.5 (Fig. 8a). The total number of EdU+ cells did not differ between the groups ($183 ± 29$ cells/mm² [experimental diet] *vs.* $301 ± 85$ cells/mm² [standard diet]; $n = 3$/group; Supplementary Fig. 9b). In contrast, the distribution of EdU+ cells across cortical laminae, which were separated by their different cellular packing densities[35,37], were

dissimilar, with an increased density of EdU+ cells at the cortical plate-subplate boundary (*layer 5*: $15 ± 1$ cells/mm² [experimental diet] *vs.* $7 ± 0$ cells/mm² [standard diet]; $p = 0.003$), as well as the subventricular zone (*layer 9*: $18 ± 1$ cells/mm² [experimental diet] *vs.* $12 ± 2$ cells/mm² [standard diet]; $p = 0.025$; Fig. 8a). Conversely, the uppermost cortical plate/marginal zone that forms first in the outside-in hierarchy of cortical lamination[38] had less EdU+ cells (*layer 2*: $4 ± 1$ cells/mm² [experimental diet] *vs.* $12 ± 5$ cells/mm² [standard diet]; $p = 0.0008$; Fig. 8a). These data suggest the layer-specific redistribution of postmitotic neuroblasts from equivalent pools of progenitors in the brains of male fetuses when dams received the experimental diet.

Moreover, we observed an ~45% decrease in the expression of growth-associated protein 43 (GAP43; $n = 3$ male mice/group; $p = 0.018$ *vs.* [standard diet]; Student's *t*-test), and ~65% reduction in microtubule-associated protein 2 (MAP2; $n = 3$ male mice/group; $p < 0.0001$ *vs.* [standard diet]) along the fetal somatosensory cortex (Supplementary Fig. 9c, d), suggesting a potential dysregulation of cortical neurogenesis during embryonic development. This decrease was also validated by iTRAQ methodology[24] ($n = 3$ mice/group; mixed for sex), allowing the unbiased testing if protein contents were affected by the experimental diet. Among the 266 proteins whose levels changed in the fetal cortices at E18.5 (129 *vs.* 137 proteins being up or downregulated, respectively; Supplementary Table 2), both GAP43 and MAP2 were found reduced in the brains of male fetuses in a gene ontology cluster identified as "*structural activity/cytoskeleton*" (Fig. 8b, c). Moreover, we found 18 proteins (7%) of the cluster "*proteasome*" and 48 proteins (18%) of "*translation, ribosome assembly, endoplasmic reticulum*" changed (Fig. 8b), providing faithful coverage of the proteostasis network[39]. Within the "*proteasome*" cluster, CaCyBp was particularly reduced (Fig. 8c; male mice) and correlated with increased S100A6 levels. Western blotting corroborated the iTRAQ data (Fig. 8d; $n = 3$ male mice/group; $p = 0.003$ *vs.* [standard diet, CaCyBp]; $p < 0.0001$ *vs.* [standard diet; S100A6]; Student's *t*-test). We have also detected reduced total protein content in fetal cortices (Fig. 8e; $n = 3$ male mice/group; $p = 0.01$ *vs.* [standard diet]), and increased tubulin acetylation (Fig. 8f; $n = 3$ male mice/group; $p = 0.002$ *vs.* [standard diet]) after gestational exposure to the experimental diet. Cumulatively, these data suggest that S100A6-CaCyBp signaling is sensitive to manipulation of dietary PUFA levels during pregnancy.

Finally, we have assessed if key parameters of cortical reorganization endure into the adulthood of male offspring prenatally exposed to an experimental diet. We particularly focused on anxiety-like behaviors because of the prevalence of this endophenotype upon manipulations with dietary PUFAs[35]. In the elevated plus-maze paradigms, we detected a significant reduction of entries in the open arms (Fig. 8g; $p = 0.023$). These data emphasize the enduring impact of PUFA manipulation on corticogenesis in vivo[23], with a life-long manifestation of depression-like behaviors in adult male offspring.

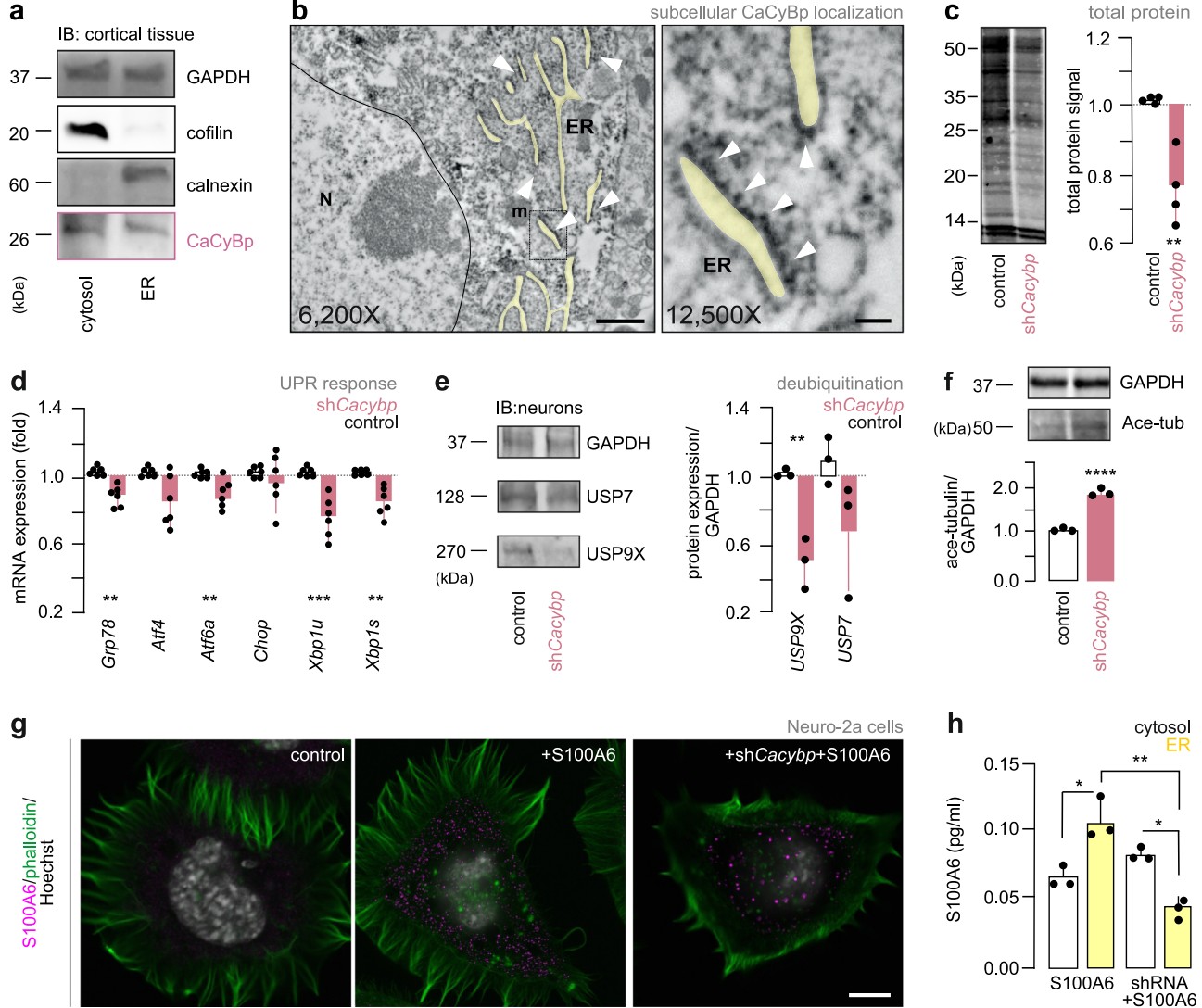

**Fig. 5 | CaCyBp modulates protein turnover in neurons. a** Identification of CaCyBp in both cytosol and endoplasmic reticulum (ER) fractions by Western blotting (WB). Cofilin and calnexin in the cytosol *vs.* ER fraction showcase sub-cellular enrichment (cortex from $n = 3$ male mice). **b** CaCyBp accumulated around the ER (*white arrowheads*) in hippocampal neurons. *Scale bars* = 500 nm (12,500×) and 1 μm (6200×) ($n = 3$ male mice). **c** WB of total protein in cortical neurons after sh*Cacybp* exposure ($n = 4$ male mice). **d** mRNA levels for *Grp78, Atf4, Atf6a, Chop* ($p = 0.700$), *Xbp1u*, and *Xbp1s* in cortical neurons after sh*Cacybp* application. Data were normalized to *Gapdh* ($n = 6$ biological replicates). **e** WB detection of ubiquitin-specific protease 7 (USP7; 128 kDa) and ubiquitin-specific protease 9× (USP9X; 270 kDa, $n = 3$ biological replicates) in cortical neurons exposed to sh*Cacybp*. Data

were normalized to GAPDH. **f** WB detection of acetylated tubulin (50 kDa) in cortical neurons exposed to sh*Cacybp* ($n = 3$ biological replicates). Data were normalized to GAPDH. **g** Neuro-2a cells exposed to sh*Cacybp*, followed by His-S100A6 (1 μg/μl) for 1 h, and double-labelled for F-actin (phalloidin; green), anti-S100A6 (magenta), and DAPI (nuclear counterstain; white). *Scale bar* = 8 μm. **h** S100A6 ELISA in Neuro-2a cytosol and ER fraction upon treatment ($n = 3$ biological repli-cates). Data were expressed as means ± s.d. and statistically evaluated by two-tailed Student's $t$-test (*$p < 0.05$, **$p < 0.01$, ***$p < 0.001$, (****$p < 0.0001$) except for (**h**), which was analyzed by one-way ANOVA. Raw data, as well as exact statistical parameters, are available in the Source Data File.

## Discussion

Astrocytes and neurons communicate through an array of funda-mental mechanisms during brain development, each playing a crucial role in forming a functional nervous system[40]. As a prototypical sig-nalling axis, brain-derived neurotrophic factor (BDNF), alike other neurotrophins and growth factors, is known to be released by astro-cytes to support neuronal survival, growth, and differentiation[1,41]. Ca²⁺ signalling is another essential mechanism, where astrocytes use pro-tracted Ca²⁺ rises to release, e.g., ATP/adenosine, modulating both neuronal activity and synaptic transmission[42].

Among the numerous signalling molecules involved, the S100 protein family plays a significant role[43]. These Ca²⁺-binding proteins can act as sensors (that is, to recruit partner proteins intracellularly to initiate signalling cascades) to modulate, e.g., cell proliferation,

differentiation, and response to injury[44]. Specifically, S100A6/calcyclin influences neurodevelopment by regulating cytoskeletal dynamics and cell survival[18,45,46]. Through these actions, S100A6, when focally released by, among other cells, astrocytes in an activity-dependent fashion, might facilitate the proper alignment, synaptogenesis, and wiring of neurons, ensuring the accuracy of connectivity in the developing brain.

Here, we described a molecular signaling pathway priming astrocyte-to-neuron interactions. Stimulation of astrocytes can lead to S100A6 release, at least in vitro. S100A6 then could bind CaCyBp intracellularly. Our study is limited in addressing if and how free S100A6 could be internalized, which is likely given that CaCyBp knock-down did not curtail intracellular S100A6 accumulation. Yet, we note that CyCyBp is not a cell-surface receptor but a sensor protein

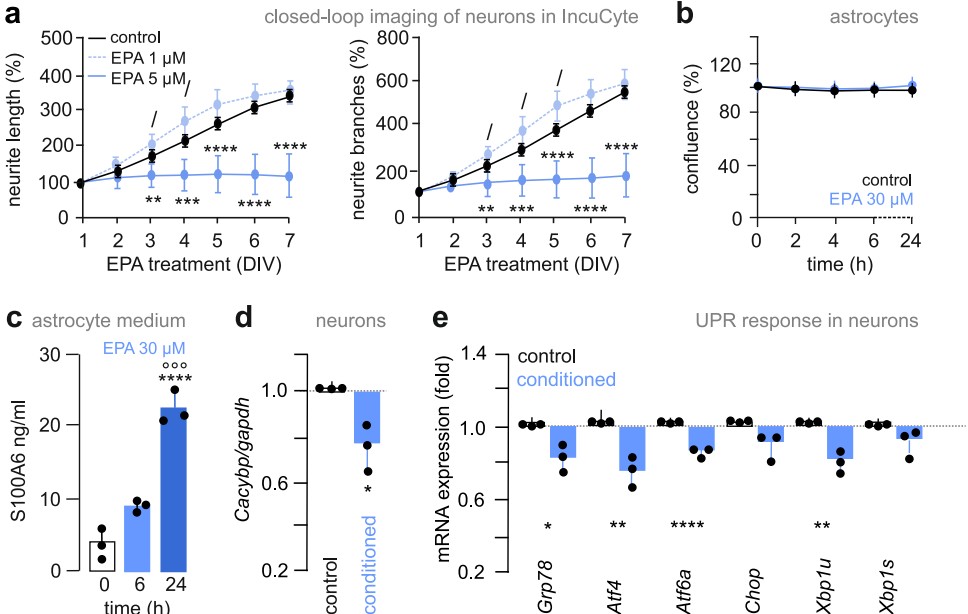

**Fig. 6 | S100A6 signaling is modulated by eicosapentaenoic acid.**
**a** Neuritogenesis in cortical neurons (prepared at E18.5) treated with EPA at either 1 μM or 5 μM concentration, and monitored by dynamic live-cell imaging for 7 days in vitro (DIV). Data were normalized to neurite length prior to drug exposure on 1 DIV. Neurite outgrowth was determined by using a manual overlay for each neuron ($n = 7$ independent experiments). **b** Cultured mouse astrocytes were stimulated with EPA (30 μM) *vs.* control. Data were normalized to cell density prior to EPA exposure on 1 DIV ($n = 8$ independent experiments). **c** ELISA of S100A6 in culture media upon EPA (30 μM) applied for 24 h ($n = 3$ biological replicates). **d** mRNA

levels for *Cacybp* in cortical neurons after exposure to a medium pre-conditioned for 24 h ($n = 3$ biological replicates). **e** mRNA levels for *Grp78*, *Atf4*, *Atf6a*, *Chop* ($p = 0.067$), *Xbp1u*, and *Xbp1s* ($p = 0.122$) in E18.5 cortical neurons after exposure to astrocyte-conditioned medium for 24 h ($n = 3$ biological replicates). Mixed-sex embryos were used where relevant. Data were expressed as means ± s.d. throughout. Data were statistically evaluated using ANOVA (**a**–**c**) or two-tailed Student's *t*-test (**d**, **e**), with only post-hoc comparisons indicated ($^{\prime}p < 0.05$, $^{*}p < 0.05$, $^{**}p < 0.01$, $^{***}p < 0.001$, $^{****}p < 0.0001$. Unprocessed data and statistics were included in the Source Data File.

adjoining the ER, and henceforth transport and targeting mechanisms ought to exist for S100A6 to pass through the plasma membrane and to become enriched in the cytosol, respectively. Notwithstanding, an ensuing shift in the activity of the ubiquitin-proteasome system (UPS), a pathway fundamental for protein degradation, likely alters cellular processes that shall operate at the peak of their activity during neuronal growth and development[19]. Particularly, slowed protein turnover could then reduce cytoskeletal instability, leading to, e.g., reduced neuritogenesis.

S100A6-CaCyBp signaling appears to be a molecular pathway sensitive to eicosapentaenoic acid (EPA), a key ω-3 fatty acid that has anti-inflammatory and neuroprotective properties[27,28,31,47]. During pregnancy, ω-3 fatty acids play a crucial role in fetal brain development[48,49]. In particular, EPA aids the formation of neuronal membranes, enhances membrane fluidity, and ensures the efficient transmission of neural signals[47]. EPA also influences gene expression to facilitate the growth and differentiation of neurons, including synaptogenesis, and myelination[50,51]. Thus, ensuring adequate ω-3 fatty acid intake during pregnancy, including EPA, is essential for physiologically adequate fetal brain development, being the foundation for cognitive and neurological health throughout life. We found that a hypercaloric experimental diet enriched in ω-3 fatty acids altered cortical architecture in male mice (that is, the positioning and molecular make-up of neurons). This phenotype was associated with an increase in S100A6, and a concomitant reduction in CaCyBp, corroborating in vitro results. These changes could also be linked to an altered status of the UPS, alike cells either treated with S100A6 or CaCyBp silenced by RNAi. It is of note that many cytoarchitectural changes seen at late-gestation became alleviated during postnatal life, yet an anxiety-like phenotype remained. This is not entirely unexpected since polyunsaturated fatty acid (PUFA) imbalance, whether due to the dominance of ω-6 or ω-3 PUFAs[35], can

affect exploratory behaviors in stressful environments, at least in laboratory rodents. Thus, we suggest that EPA-enriched diets can influence fetal brain development through the modulation of modes of intercellular communication, potentially imposing long-lasting modifications that can compromise resilience to environmental challenges later in postnatal life.

In conclusion, our results describe a molecular axis, which limits neuronal morphogenesis through modulating the ER/UPS pathway in mouse brain. By using S100A6 as a released ligand, astrocytes, among other cells, could define endpoints of neuritogenesis through a signaling cascade centered on the regulation of proteostasis.

## Methods

### Animals and ethical approval of in vivo studies
Female mice (C57Bl6/J) were housed in groups (5 mice/group) in perplex cages on a 12 h/12 h light/dark cycle (lights on at 08:00 h) and in a temperature ($22 \pm 2$ °C) and humidity ($50 \pm 10$%)-controlled environment. Food and water were available *ad libitum*. Embryos and tissues were obtained from timed matings with the day of vaginal plug considered as embryonic day (E) 0.5. Group sizes were defined as per conventions in developmental biology[35,52]. Experiments on live animals conformed to the 2010/63/EU European Communities Council Directive and were approved by the Austrian Ministry of Science and Research (66.009/0277-WF/V/3b/2017). Particular effort was directed towards minimizing the number of animals used and their suffering during experiments. Isoflurane (1–5%/L/min air flow, as appropriate) was used as anesthesia in all experiments.

### Single-cell RNA-sequencing data analysis
We analyzed single-cell RNA-sequencing data from developing mouse cortex spanning embryonic day (E) 10 to postnatal day (P) 4. The dataset was obtained from Ref.[6], and accessed through the Single Cell

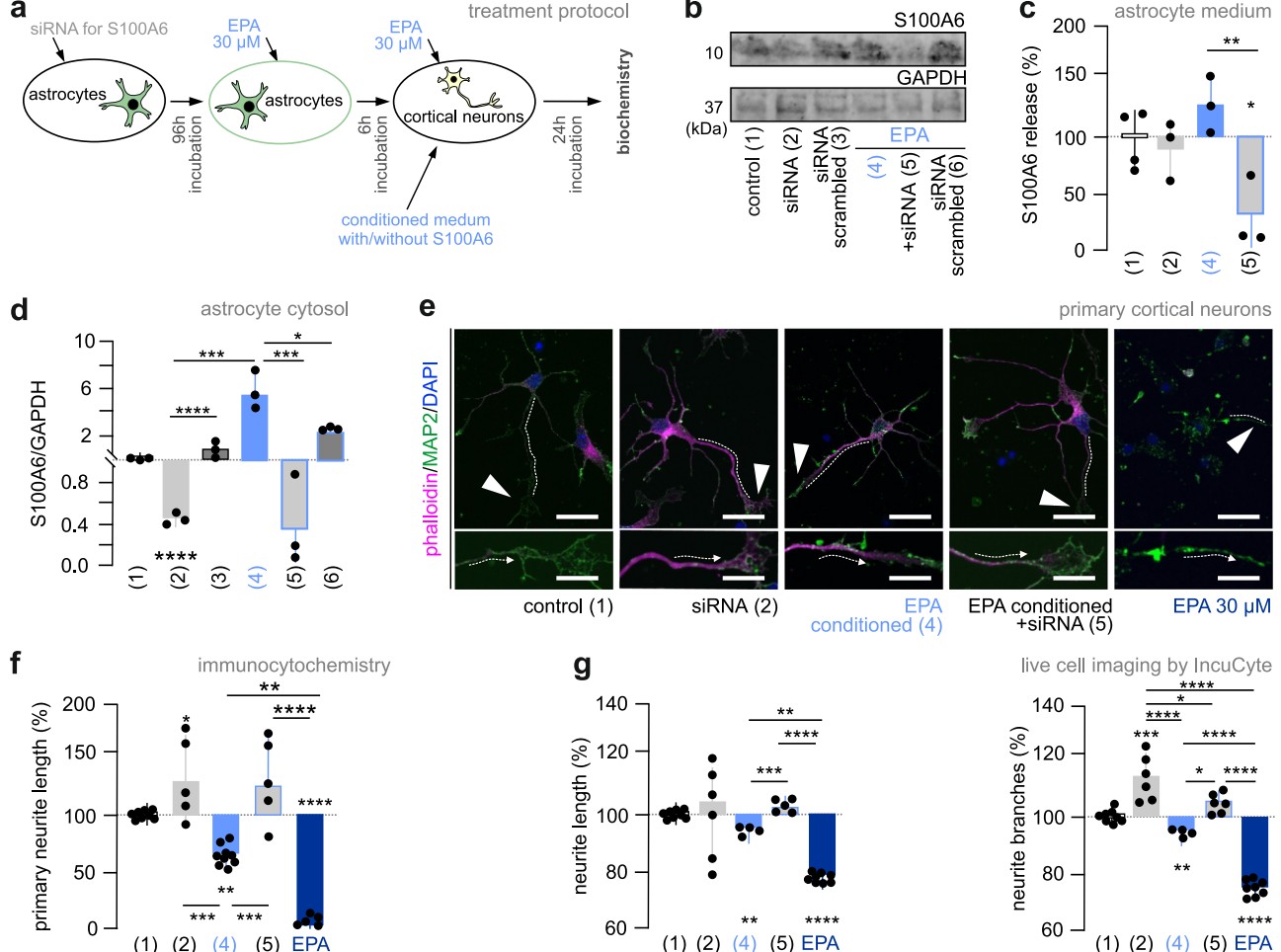

**Fig. 7 | S100A6 availability modulates neuronal differentiation. a** Co-culture experiments schema. **b** S100A6 (10 kDa) WB in astrocyte cytosol after 96 h exposure to scrambled or targeting siRNA, followed by 6 h EPA (30 μM) treatment. Representative blots are shown; lane numbers correspond to subsequent panels for clarity. **c** S100A6 ELISA in culture media upon treatment as in (**b**) with either $n = 4$ biological replicates [control] or $n = 3$ biological replicates [siRNA; EPA; EPA + siRNA]. **d** S100A6 detection in the cytosol of astrocytes after treatments as in (**b**) ($n = 3$ biological replicates). **e** Representative imagines of cortical neurons exposed to conditioned media for 24 h or EPA (30 μM), and double-labeled for F-actin (phalloidin; magenta) and MAP2 (green). Lead neurites and growth cones were marked by dashed lines and solid *arrowheads*, respectively. *Scale bars* = 12 μm (*top*),

5 μm (*bottom*). **f** Primary neurite length after experimental manipulations (biological replicates: $n = 7$ [control]; $n = 5$ [cond. siRNA; cond. siRNA + EPA; EPA]; $n = 9$ [cond. EPA]. **g** Neuritogenesis after genetic and pharmacological manipulations and monitored for 24 h. Data were normalized to neurite length prior to treatment and expressed in percentage. Neurite outgrowth was determined by using a manual overlay (biological replicates: $n = 7$ [control; cond. siRNA], $n = 4$ [cond. EPA], $n = 5$ [cond. siRNA + EPA], $n = 8$ [EPA]). Data were expressed as means ± s.d. throughout. Samples were mixed for sex where relevant. Data were statistically evaluated by one-way ANOVA followed by Tukey's multiple comparisons with post-hoc comparisons indicated (*$p < 0.05$, **$p < 0.01$, ***$p < 0.001$, ****$p < 0.0001$. Unprocessed data and detailed statistics were included in the Source Data File.

Portal (SCP1290)[53]. Raw count data and metadata were downloaded and processed using Seurat (v5.1.0) in R (version 4.4.0 (24/04/2024))[54,55]. We chose Seurat for its comprehensive toolset for quality control, analysis, and exploration of single-cell RNA-seq data, as well as its wide adoption in the field.

**Data preprocessing and quality control.** The raw count matrix was loaded using the "*Read10X function*" in Seurat. We performed the following pre-processing steps: (*i*) the log1p normalized matrix was converted back to raw counts by applying "*expm1*"; (*ii*) scaling factors were calculated based on the total UMI counts/cell. The count matrix was scaled by multiplying each cell's counts by its scaling factor; (*iii*) a new Seurat object was created using the scaled count matrix; (*iv*) cells annotated as doublets, low quality, or red blood cells were removed using the "*subset*" function; (*v*) data were then normalized using the "*NormalizeData*" function, and 5000 highly variable features were identified using "*FindVariableFeatures*".

**Dimensionality reduction and clustering.** We performed principal component analysis (PCA) on variable features using "*RunPCA*". Based on the Elbow plot, which indicates the variability of each principal component (PC), we selected the first 30 out of 50 PCs for downstream analysis. This choice helped us to reduce noise in the data while ensuring biological reproducibility. Uniform Manifold Approximation and Projection (UMAP)[56] and *t*-distributed Stochastic Neighbor Embedding (t-SNE)[57,58] were used for dimensionality reduction, with embeddings stored in the Seurat object. Both techniques used the selected 30 PCs as input. Cells were clustered using the "*FindNeighbors*" and "*FindClusters*" functions. For community detection, we employed the Leiden algorithm (resolution = 0.7) instead of the commonly used Louvain algorithm or alternatives such as walktrap, multilevel, or infomap. The Leiden algorithm was chosen for its ability to find converged optimal solutions more efficiently, which is particularly beneficial for large-scale single-cell datasets[59].

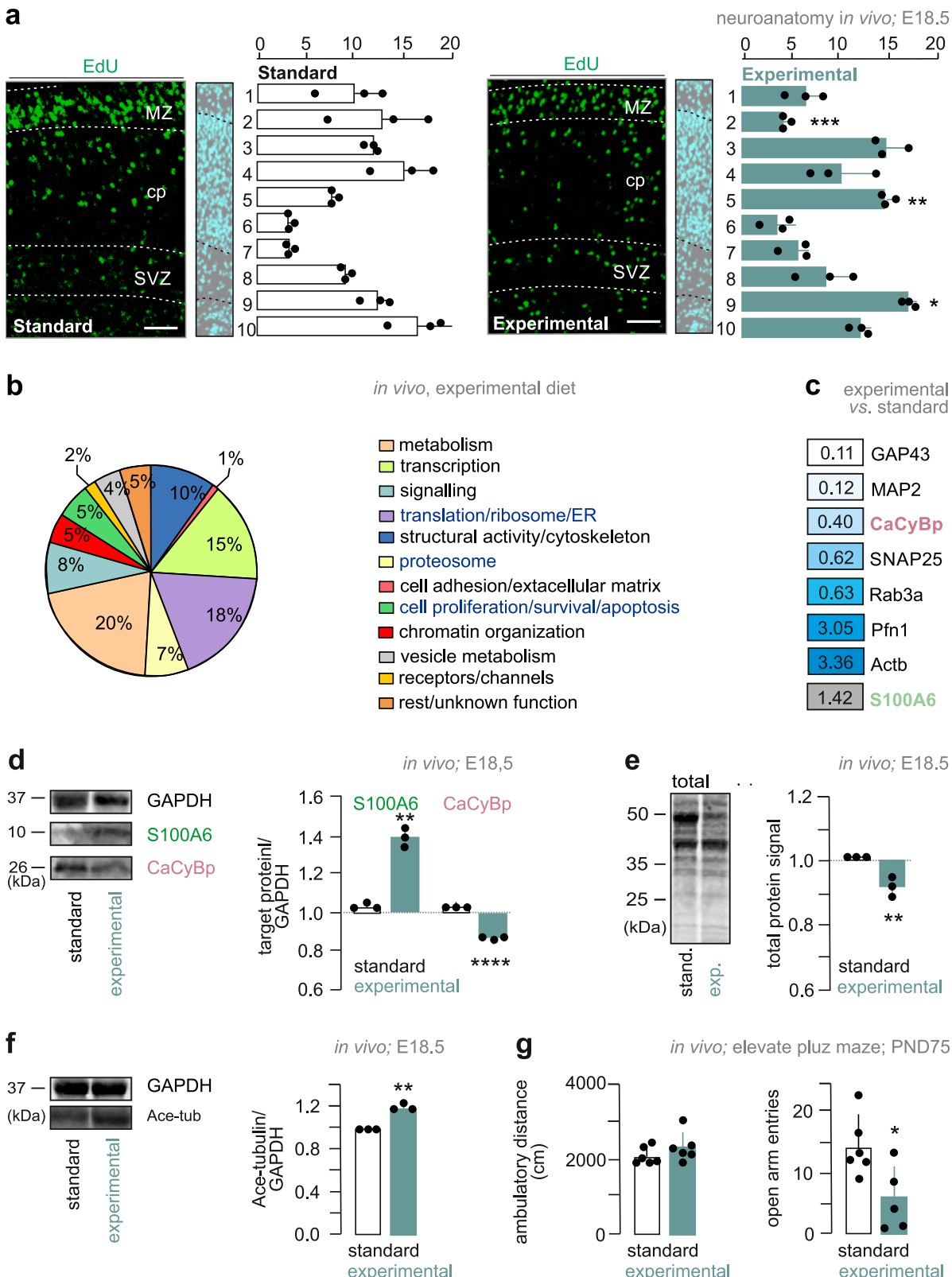

**Gene expression analysis.** We analyzed the expression of S100 family genes and a curated list of genes of interest across different developmental stages and cell types. Feature plots, violin plots, and dot plots were generated using Seurat's visualization functions ("*FeaturePlot*", "*VlnPlot*", "*DotPlot*") and custom functions from the scCustomize package (v2.1.2).

**Analysis of the astrocyte lineage.** Cells annotated as astrocytes, apical progenitors, and cycling glial cells were subset for focused analysis of the astrocyte lineage. This subset was re-clustered using the same approach as above. We performed differential expression analysis between astrocyte clusters using both the "*FindAllMarkers*" function in Seurat (using a logistic regression test[60]) and DESeq2

**Fig. 8 | Transgenerational modulation of S100A6 signaling. a** Representative images (*left*) and quantification (*right*) of EdU⁺ neurons in cortical layers of male embryos at E18.5. Cell counting was performed in binned layers (numbered consecutively from 1 to 10) on equal tissue surfaces, and expressed as absolute numbers ($n = 3$ biological replicate; mixed for sex; *layer* 2: ***$p < 0.001$; *layer* 5: **$p = 0.003$; *layer* 9: *$p = 0.025$). *Scale bars = 20 μm.* cp cortical plate, MZ marginal zone, SVZ subventricular zone. **b** Classification of significantly altered proteins in mixed sex embryos through primary function assignment by gene ontology. Dams were exposed to a hypercaloric diet enriched in ω−3 fatty acids (Supplementary Table 1). **c** Topmost modified proteins targets, with CaCyBp significantly downregulated after gestation experimental diet exposure. In contrast, S100A6 was upregulated. Only proteins contributing to the morphological differentiation of neurons were included (*see also* Supplementary Table 2). **d** Western analysis of

S100A6 and CaCyBp protein levels ($n = 3$ male mice/group; ****$p < 0.0001$, **$p = 0.003$ *vs.* standard diet) in the cerebri of male embryos on E18.5. Data were normalized to GAPDH. **e** Total protein content of cerebral tissues from male embryos at E18.5 ($n = 3$ male mice/group; **$p = 0.001$ *vs.* standard diet). **f** Western analysis of acetylated tubulin (50 kDa) in the cortices of male embryos at E18.5 in control and after exposure to an experimental diet ($n = 3$ male mice/group; **$p = 0.002$ vs. standard diet). Data were normalized to GAPDH. **g** Elevated plus maze test results expressed as "distance travelled" (cm) and "number of arm entries" ($n$) from $n = 6$ adult male mice/control group and $n = 5$ adult male mice/experimental diet group (*$p = 0.023$ *vs.* standard diet). Data were expressed as means ± s.d. throughout, and statistically evaluated using either two-way ANOVA (**a**) or two-tailed Student's *t*-test (**d**–**g**). Unprocessed data and statistics were included in the Source Data File.

(v1.44.0)[61] on pseudo-bulk data aggregated by cluster and developmental stage. The combination of these two approaches allowed us to leverage the strengths of both single-cell and bulk RNA-seq differential expression methods. Data preprocessing, quality control (including droplet selection, ambient RNA removal, doublet detection, and refined filtering), clustering, marker gene definitions, and the specific software packages used for data processing were made available at https://harkany-lab.github.io/Cinquina_2024/cortex_visualisation.html.

### Human tissue, immunohistochemistry
Fetal brains (gestational week (GW) 27; $n = 3$ samples; mixed for sex) were collected at the Neurobiobank of the Institute of Neurology, Medical University of Vienna, Austria. Tissues were obtained and used in compliance with the Declaration of Helsinki and following institutional guidelines approved by the Ethical Committee of the Medical University of Vienna (IRB No. 396/2011). Fetal brain tissue was obtained from spontaneous or medically induced abortions with informed consent from the mothers. Fetuses whose cause of death was unrelated to genetic disorders, head injury, neurological diseases, or other known diseases (e.g., infections) were only included. Tissues were kept at 4 °C for 24–48 h due to local regulations before immersion fixation in formalin (10%) on average for 34–37 days before embedding in paraffin. Three-μm tissue sections of formalin fixed, paraffin-embedded tissue blocks were mounted on pre-coated glass slides (StarFrost). Following deparaffinization and rehydration, sections were incubated in a solution of $H_2O_2$ (3%) and MeOH (Millipore; #67_56_1) for 10 min to block endogenous peroxidase activity. After washing with distilled water, sections were pre-incubated in low pH EnVision™ FLEX antigen retrieval solution (PTLink; Dako) at 98 °C for 20 min, exposed to fetal calf serum (10%) for 10 min and subsequently stained with rabbit anti-CaCyBp antibody (Supplementary Table 3) at 4 °C for 72 h. The EnVision detection kit (K5007, ready-to-use, DAKO) was used to visualize the horseradish peroxidase/3,3'-diaminobenzidine (DAB) reaction product with $H_2O_2$ substrate (0.01%). Sections were counterstained with hematoxylin-eosin, dehydrated in an ascending series of ethanol, cleared with xylene, covered with Consil-Mount (Thermo Scientific), and viewed under a light microscope (Nikon Eclipse E400). We did not observe variations in the cellular distribution of CaCyBp across the samples. Therefore, a representative image was included.

### Immunohistochemistry, light and electron microscopy in mice
Adult male mice were transcardially perfused with ice-cold 0.1 M PB (20 ml), followed by 4% PFA in PB (100 ml at 3 mL/min flow speed). P120 adult brains were post-fixed in 4% PFA overnight. After equilibrating in 30% sucrose for 48–72 h, coronal serial sections (1-in-6 series) were cryosectioned (Leica CM1850) at a thickness of 50 μm. Free-floating sections were processed as described[62] for either fluorescence microscopy or chromogenic (using DAB as substrate) electron

microscopy. E18.5 and P4 brain were immersion fixed in 4% PFA overnight, rinsed in PB, and cryoprotected in 30% sucrose (in PB) for 48 h. Serial coronal sections (20 μm) were cut, thaw-mounted onto fluorescence-free SuperFrost⁺ glass slides, and stored at −20 °C until processing.

For fluorescence histochemistry, E18.5, P4, and P120 sections from $n = 4$ male mice were extensively rinsed in PB, and blocked with PB containing 5% normal donkey serum (NDS; Jackson ImmunoResearch; #017000121), 1% bovine serum albumin (BSA; Sigma; #A2153), and 0.3% Triton ×-100 (Sigma) at 21–24 °C for 2 h to reduce non-specific labeling. Next, sections were incubated with a combination of primary antibodies (Supplementary Table 3) diluted in PB also containing 2% NDS, 0.1% BSA, and 0.3% Triton ×-100 at 4 °C for 72 h. DyLight Fluor 488, 560, or 633-tagged secondary antibodies (1:300; Jackson ImmunoResearch) were used for signal detection. Hoechst 33,342 (Sigma; #B2261) and 4',6-diamin-2-phenylindol (DAPI, Sigma; #D8417) were routinely used as nuclear counterstains. After washing in PB and a final rinse in distilled water, free-floating sections were mounted onto fluorescence-free glass slides and coverslipped with Mowiol mounting medium (Sigma; #81381). Fluorescence images were acquired on a Zeiss 880 LSM confocal laser scanning microscope equipped for maximal signal separation and spectral scanning/unmixing.

Electron microscopy was performed as previously published[23]. Briefly, free-floating sections from male adult mice were washed three times with PB for 5 min each and then treated with 1% $H_2O_2$ (Sigma; #216763) for 10 min to block endogenous peroxidase activity. After rinsing, sections were exposed to a blocking solution (*see above*) at 21–24 °C for 2 h. Sections were then incubated with a rabbit anti-CaCyBp primary antibody (Supplementary Table 3) for 72 h. After washing three times, sections were exposed to biotinylated anti-rabbit secondary antibodies (1:200; Jackson ImmunoResearch, 20–22 °C, 2 h), rinsed again in PB, and then exposed to preformed avidin-biotin-peroxidase complexes (1:200, ABC Elite; Vector Laboratories; #PK6100) at 4 °C overnight. Next, biotin-labeled cells were revealed using a peroxidase reaction with DAB alone (5 mg/ml, 0.03% $H_2O_2$ as substrate, brown reaction product). Sections were subsequently osmificated (1% $OsO_4$ in PB, 15 min), dehydrated in an ascending ethanol gradient followed by flat embedding in Durcupan (FLUKA, ACM). Ultrathin sections (60 nm) were cut on a Leica ultramicrotome, collected on Formvar-coated single slot grids, and analyzed with a Tecnai 10 electron microscope (FEI; primary magnifications: 6200×; 17500×).

### In situ hybridization (HCR-FISH)
Fresh-frozen E18.5 and P4 mouse brains were cryosectioned at a thickness of 16 μm and collected on SuperFrost⁺ glass slides (ThermoFisher). Sections were immersed in 4% PFA for 15 min, and dehydrated in an ascending ethanol gradient (50%, 70%, and 100%; 5 min each). The HCR™ RNA-FISH protocol for either fresh-frozen or fixed-frozen tissue sections (Molecular Instruments) was used with pre-

designed *Cacybp, S100a6, Rbfox3, Gfap, S100b, Aldh1l1,* and *Apo4* probes. Samples were imaged on an LSM 880 confocal microscope (Zeiss) at 40× primary magnification.

### Neuronal cultures, shRNA knockdown, treatment with recombinant S100A6

Cerebral tissues (cortex/hippocampus) of mouse embryos (mixed for sex) were isolated on either E18.5 or P0. Cells were mechanically dissociated into a single-cell suspension by papain digestion (20 U/ml, pH 7.4, 45 min; Worthington, #LK003176) and plated onto poly-D-lysine-coated (PDL; Sigma; #P728) 6-well (200,000 cells/well) or 24-well (50,000 cells/well) plates. Cells were maintained in Neurobasal medium (Fisher Scientific; #10888_022) supplemented with B27 (1%; Thermo Fisher; #17504-044), GlutaMAX™ (1%; ThermoFisher; #35050_038), and penicillin/streptomycin (1%; Life Technologies; #15140_122). Media were replaced every other day.

For shRNA knock-down of *Cacybp*, P0 cortical neurons (200,000 cells/well) were seeded in 6-well plates, grown for 48 h, then transduced with either 1 µl of adeno-associated virus, serotype 2 (AVV-2) particles (6.09 × 10$^{12}$ genome copies/ml; GeneCopoeia™; #AA02-CS-MSE091282-AVE001-01-200) to carry shRNA or equal amounts of AVV-2s to deliver scrambled controls (GeneCopoeia™; #AA02-CS-CSECTR001-AVE001-01-200). Media were replaced with Neurobasal medium (*as above*) free of AAV-2 particles after 48 h, and changed every other day. Alternatively, P0 neurons in 6-well plates were exposed to either 1 µg of recombinant His-tagged S100A6 protein (Abcam; #ab108124) or 1 µg of recombinant His-tagged S100A4 protein (Abcam; #ab109341) for 24 h on days in vitro (DIV) 14. Analyses of neuronal morphology were performed on DIV15 because *Cacybp* mRNA and protein levels were found significantly reduced by then in shRNA-exposed samples *vs.* both technical controls and samples treated with scrambled shRNA. Cells were then lysed for Western blotting and qPCR after DIV15. Morphometric analysis of cultured neurons was aided by the ImageJ software and included the length of the primary neurite (µm).

In combined experiments, P0 neurons were plated in 6-well plates (200,000 cells/well), while E18.5 neurons in glass bottom dishes (25,000 cells/well; 35 mm diameter, 20 mm well size; 0.13−0.16 mm bottom; Cellvis, #D35-20-1-N). Cells were grown for 48 h, then transfected with AVV-2 particles as above. After 14 DIV, neurons were exposed to 1 µg of recombinant His-tagged S100A6 protein for 24 h. Neurons in 6-well plates were then processed for RNA extraction and qPCR. Neurons in glass bottom dishes were immersion-fixed in 4% PFA (in 0.1 M phosphate-buffered saline [PBS]) for 1 h, and then exposed to a blocking solution consisting of 10% NDS, 5% BSA, and 0.3% Triton ×-100 in PBS (pH 7.4) at 21−24 °C for 2 h. To visualize F-actin, cultured neurons were incubated with Alexa Flour 546-conjugated phalloidin (1:500; Invitrogen; Supplementary Table 3) for 2 h. Hoechst 33,342 (Sigma; #B2261) was routinely used as nuclear counterstain.

To test the effects of astrocytes (with/without genetic manipulation; *see below*), E18.5 neurons were plated first in 24-well plates (50,000 cells/well), 96-well plates (20,000 cells/well for morphometry), or 6-well plates (1 × 10$^6$ cells/well for qPCR), and allowed to grow until 3 DIV. At this point, half of the neuronal media was replaced with either fresh neuronal media or astrocyte-conditioned media for another 24 h. Neuronal morphology and gene expression were then analyzed.

For S100A4 treatment, E18.5 neurons were plated at a density of 50,000 cells/well on glass bottom dishes (*as above*). After 3 DIV, neurons were exposed to 1 µg of recombinant His-tagged S100A4 protein (Abcam; #ab109341) for 24 h. Cells were then immersion-fixed in 4% PFA, and incubated in a blocking solution for 2 h (*as above*). Subsequently, samples were exposed to a guinea pig anti-MAP2 primary antibody (Supplementary Table 3) diluted in PBS also containing 2% NDS, 0.1% BSA and 0.3% Triton ×-100 at 4 °C for 24 h. DyLight Fluor 488-tagged secondary antibody were used to visualize MAP2 localization.

After rinsing the samples in PBS and finally in distilled water, coverslips were mounted onto fluorescence-free glass slides using Mowiol. Images were acquired on a Zeiss 880LSM confocal laser scanning microscope with maximal signal separation and spectral scanning/unmixing.

### High-throughput neurite tracking

Mouse cerebral cortices were isolated on E18.5. Cells were mechanically dissociated and plated at a density of 20,000 cells/well in PDL-coated 96-well plates for on-line morphometry. After 24 h, E18.5 cultures were exposed to either recombinant S100A6 (1 µg/µl; Abcam #ab108124) for 4 DIV or EPA (either 1 µM or 5 µM; Cayman, #90165) for 7 DIV. Before use, EPA was flushed with N$_2$, then dissolved in DMSO. Aliquots were stored in glass amber vials at −20 °C until needed. Working concentrations of the EPA stock solution were diluted as appropriated to achieve the necessary final concentration in neuronal media containing BSA as antioxidant. Treatment was in quadruplicate and imaged live using an IncuCyte Zoom high-throughput device (Sartorius). Time-lapse images were acquired in loops with 2-h intervals. The growth rate of neurites in each well was obtained by measuring the surface area covered by neurites and expressed as mm/mm$^2$ surface area. Surface occupancy of the neurons after either 4 DIV or 7 DIV in culture was computed as the percentage surface area covered by live neurons ("*body cluster area*"), and taken as a measure of cell survival.

### Primary cultures of astrocytes, confluence analysis, and S100A6 secretion

Cortices and hippocampi of P7 mice (*n* = 7) were isolated, dissociated into single cell suspension by papain (*as above*), and plated in T75 flasks. Cells were maintained in DMEM (high glucose, 1% GlutaMAX™; Thermo Fisher, #61965_026) supplemented with Na-pyruvate (1%; Sigma #11360_070), fetal bovine serum (FBS; 10%, Sigma, #A5256801), and 1% penicillin/streptomycin. After 7−9 days, T75 flasks were placed on a shaker (180 rpm, 37 °C, overnight) to remove cell debris, neurons, and oligodendrocytes.

For confluence analyses, astrocytes were plated at a density of 7000 cells/well into 96-well plates. After 24 h, astrocytes were placed in serum-free medium for 24 h, and then treated with glutamate (100 µM; Tocris, #56_86_0) or EPA (5, 10, and 30 µM; Cayman, #90110) for another 24 h. Before use, glutamate was dissolved in sodium hydroxide. EPA was flushed with N$_2$, dissolved in dimethyl sulfoxide, its aliquots stored at −20 °C, and used as above. Both glutamate and EPA treatments were performed in quadruplicate and imaged live using an IncuCyte Zoom high-throughput imaging device (Sartorius). Time-lapse images were acquired every 2 h. Confluence was determined as above.

To detect S100A6 secretion, astrocytes (1 × 10$^6$) were trypsinated (0.25% trypsin-EDTA, Gibco, #25200_56), and seeded in 6-well plates. After 24 h, astrocytes were starved in serum-free medium at 37 °C for 24 h, and then treated with either glutamate (100 µM) or EPA (5, 10, and 30 µM) for an additional 24 h. Supernatants were collected and concentrated using either Amicon Ultra-4 centrifugal units (3 kDa cut-off; Sigma, #UFC800324) by centrifugation at 7500 × *g* for 40 min or Amicon Ultra-0.5 centrifugal units (3 kDa cut-off; Sigma #UFC500396) by centrifugation at 14,000 × *g* for 30 min. Concentrated supernatant was then used to detect S100A6 by Western blotting and/or an ultra-sensitive mouse anti-S100A6 ELISA (antibodies-online.com; ABIN425090)[63] as per the manufacturer's protocol. Astrocytes were pelleted, lysed, and used for Western blotting.

### siRNA-mediated knockdown, benchmarking

To knock-down S100A6 expression, astrocytes were plated at a density of 300,000 cells/well in 12-well plates. After 24 h, astrocytes were placed in serum-free medium for 24 h. S100A6 expression was

silenced by siRNA (1 μM; Dharmacon, #SO-3248729G). Scrambled siRNA (1 μM; Dharmacon, #D-001950-01-50) was used as control. Cells were lysed and collected for Western blotting after 96 h. To benchmark this manipulation, S100A6 protein was shown to be significantly reduced in siRNA-exposed samples *vs.* both technical controls and samples treated with scrambled siRNA (Fig. 7). After RNAi-mediated S100A6 silencing, astrocytes were treated with EPA (30 μM) for 6 h, when their pellets and supernatants were collected for Western blotting.

## MTT assay
The 3-(4,5-dimethylthiazol-2-yl)−2,5-diphenyltetrazolium bromide colorimetric assay was carried out at the end of each IncuCyte run to determine drug effects on the viability of astrocytes. MTT (Abcam; #ab146345) was added to each well at a final concentration of 0.5 mg/ml, incubated for 3 h, with the medium then aspirated. One hundred μl of 100% DMSO (Sigma; #472301) was then added to each well. After the formazan crystals had dissolved, absorbance was determined spectrophotometrically at 570 nm. Data on astrocyte viability were analyzed using "*Data analysis using Bootstrap-coupled estimation*" (DABEST package v2023.9.12) in *R* for effect sizes and their confidence intervals[64] (*see also*: https://harkany-lab.github.io/Cinquina_2024/methods.html).

## Neuro-2a cells, transfection, immunocytochemistry
Neuro-2a cells (CCL-131, ATCC) were kept in DMEM (high glucose, 1% GlutaMAX™, ThermoFisher, #61965_026) supplemented with 1% Na-pyruvate, 10% FBS, and 1% penicillin/streptomycin. Neuro-2a cells were trypsin dissociated every 4th day, and newly passaged cells were used in experiments 24−48 h after re-plating. Neuro-2a cells were directly lysed in either Laemmli buffer (for Western blotting; Bio-Rad, #1610747) or RNA extraction medium (for qPCR) or used for immunoprecipitation (*see below*).

**Immunocytochemistry on Neuro-2a cells**. Cells were plated at a density of 50,000 cells/well in 24-well plates. After 24 h, cells were exposed to recombinant S100A6 (1 μg/μl; Abcam, #ab108124;) for 1 h, immersion-fixed in 4% PFA (in 0.1 M PBS), and exposed to a blocking solution consisting of 10% NDS, 5% BSA, and 0.3% Triton ×-100 in PBS (pH 7.4) at 21−24 °C for 2 h. Next, samples were incubated with rabbit anti-S100A6 primary antibody (Supplementary Table 3) diluted in PBS also containing 2% NDS, 0.1% BSA, and 0.3% Triton ×-100 at 4 °C for 24 h. Alexa Flour 546-conjugated phalloidin (1:500; Invitrogen; Supplementary Table 3) and DyLight Fluor 488-tagged secondary antibody (1:300; Jackson ImmunoResearch) were used for F-actin and S100A6 signal detection, respectively. Hoechst 33,342 (Sigma, #B2261) was routinely used as nuclear counterstain. After washing in PBS and a final rinse in distilled water, coverslips were mounted onto fluorescence-free glass slides with Mowiol. Images revealing the subcellular localization of S100A6 were acquired on a Zeiss 880LSM confocal laser scanning microscope with maximal signal separation and spectral scanning/unmixing.

**Subcellular distribution/localization of S100A6**. Neuro-2a cells were plated at a density of either 50,000 cells/well in 24-well plates or $10 \times 10^6$ cells in 10-cm Petri dishes for 24 h. Subsequently, they were transduced with 1 μl of AVV-2 particles for shRNA delivery (*as above*). Media were replaced with DMEM free of AAV-2 particles after 48 h. After an additional 24 h, cells were exposed to recombinant S100A6 (1 μg/μl) for 1 h and then processed for either immunocytochemistry or subcellular fractionation (*see below*). S100A6 were then detected in both the cytosolic and endoplasmic reticulum fractions by ELISA (antibodies-online.com; #ABIN425090) according to the manufacturer's instructions[63].

## Immunoprecipitation and Nano LC-MS/MS measurements
Neuro-2a cells were plated at a density of $10 \times 10^6$ cells in 10-cm Petri dishes and exposed to recombinant S100A6 (1 μg/μl) for 1 h. Total protein was extracted, and immunoprecipitation performed using HisCube Ni-INDIGO His-tag MAgBeads MINI Kit according to the manufacturer's instructions. Briefly, 20 μl of Ni-beads (Cube Biotech, #75201) were transferred to 1.5 mL low-binding tubes, magnetized in a magnetic rack, with the supernatant discarded. Beads were washed (3×) with 0.5 ml PB (each washing step with short vortexing, spin-down, magnetizing, and discarding the supernatant). Proteins were added to washed Ni-beads and incubated at 4 °C overnight. Subsequently, samples were magnetized and washed (3×). Beads were eluted (5×) by adding 30 ml imidazole (500 mM, pH 7.4), incubated for 10 min on a shaker at room temperature, and magnetized. Eluents were collected and added to ice-cold acetone (1:4) and left to precipitate at −20 °C overnight. Precipitated protein samples were centrifugated at $14,000 \times g$ at 4 °C for 1 h, left to dry, and then solubilized in triethylammonium bicarbonate buffer (50 mM) on a shaker at room temperature for 2 h. Trypsin digestion was performed at 37 °C for 16 h. Formic acid was then added to the collection tubes, eluents were evaporated in a SpeedVac at 30 °C for 4 h, and processed for LC-MS.

Mass spectra were obtained using an Orbitrap Q Exactive mass spectrometer (Thermo Fisher Scientific) equipped with a nanospray ion source coupled to a HPLC system (UltiMate 3000, Dionex). Five μL of each sample was first loaded on a pre-column (trap column; Acclaim PepMap100, 100 μm × 2 cm, nanoViper C18, 5 μm, 100 Å, Thermo Scientific) for 10 min using a loading solvent (2% acetonitrile (ACN), 98% $H_2O$, 0.1% trifluoroacetic acid) at a flow rate of 10 μl/min. Separation was carried out on a C18 analytical column (Acclaim PepMap RSLC, 75 μm × 50 cm, nanoViper C18, 2 μm, 100 Å; Thermo Scientific) at a flow rate of 300 nL/min. Column oven temperature was kept at 40.0 °C. Mobile phase A composition was 2% ACN, 98% $H_2O$, 0.1% formic acid (FA), while mobile phase B was made up of 80% ACN, 20% $H_2O$, and 0.1% FA. Gradient conditions were within 135 min retention time at a flow rate of 0.3 μl/min, with mobile phase B set to 2%−90%. MS scans were acquired in positive ion mode over an *m/z* range of 400−1400 at a resolution of 70.000 (FWHM at *m/z* 200). Using a data-dependent acquisition mode, the 6 most intense precursor ions with charge states 2+ to 4+ were selected for fragmentation at 30% normalized collision energy and analysed in the Orbitrap at a resolution of 17.500 (at *m/z* 200). The dynamic exclusion for the selected ions was 30 s. The electrospray voltage was 2.2 kV and the ion transfer capillary temperature was 300 °C. Raw data were converted into mascot generic format (mgf) files using the msConvert tool from the ProteoWizard toolkit (ProteoWizard: Home (sourceforge.io)), and subsequently processed with ProteinProspector (version 6.5.0, University of California). Data were searched against N-terminally His-tagged mouse S100A6/PRA protein sequences (Accession No. P14069, MGSSHHHH HHSSGLVPRGSHMACPLDQAIGLLVAIFHKYSGKEGDKHLSKKELKELIQK ELTIGSKLQDAEIARLMDDLDRNKDQEVNFQEYVAFLGALALIYNEALK), as well as the whole mouse proteome in the SwissProt database (version 2021), which yielded identical results. As further confirmation, pure recombinant S100A6/PRA protein was analyzed as standard.

## RNA isolation and quantitative PCR
The RNeasy mini kit (Qiagen, #7326820) was used to extract RNA from mouse cerebral tissues and primary cultures. DNase I was used to eliminate traces of genomic DNA. Total RNA was then reverse transcribed to a cDNA library in a reaction mixture using a high-capacity cDNA reverse transcription kit (Applied Biosystems, #4368814). The cDNA library was then used for quantitative real-time PCR (CFX-connect, Bio-Rad). Pairs of PCR primers specific for *Cacybp, S100a6, Grp78, Atf4, Atf6a, Chop, Xbp1u, Xbp1s, Map2, Tubb3, S100b, Aldh1l1,* and *Actb* were designed with Primer Bank and the NCBI Primer Blast software (Supplementary Table 4). Quantitative analysis of gene

expression was performed with SYBR Green Master Mix Kit (Life Technologies, #1725124). Expression levels were normalized to glyceraldehyde-3-phosphate dehydrogenase (*Gapdh*), a housekeeping gene, for every sample in parallel assays ($n \geq 3$ biological replicates/group).

### Total protein determination and subcellular fractionation

Total protein was extracted from cerebral tissues and primary cultures by using a modified radioimmunoprecipitation assay buffer containing 5 mM NaF (Sigma, #7681_49_4), 5 mM $Na_3VO_4$ (Sigma, #13721_39_6), 0.1% N-octyl-β-d-glucopyranoside (Sigma, #08001), and a cocktail of protease inhibitors (Complete™, Roche, #11873580001). Samples were homogenized in a Dounce homogenizer, and protein lysates were pelleted at 10,000 *rpm* at 4 °C for 10 min. To fractionate endoplasmic reticulum (ER) and cytoplasm, samples were homogenized in 0.32 M Sucrose (Sigma, #57903); 1 mM $NaHCO_3$ (Sigma, #56297), 1 mM $MgCl_2$ (Sigma, #A0344733); 0.5 mM $CaCl_2$ (Sigma, #C7902) and a cocktail of protease inhibitors and centrifuged twice at $700 \times g$ at 4 °C for 5 min. The supernatant was centrifuged several times to collect the ER fraction ($6300 \times g$ at 4 °C for 10 min, $20,000 \times g$ for 30 min, and $100,000 \times g$ for 1 h). The final pellet was then reconstituted as the ER fraction, whereas the supernatant was saved as the cytosol fraction. Protein concentrations were determined by the Pierce™ BCA Protein Assay kit (Thermo Fisher, #23225). Cofilin and calnexin antibodies were used to control enrichment of the cytosol and ER fraction, respectively (Supplementary Table 3).

### Quantitative Western blotting, including total protein normalization

Western blotting was performed with primary antibodies listed in Supplementary Table 3. Blots were scanned on a Bio-Rad XRS$^+$ imaging system and then quantified in Image Lab 3.01 (Bio-Rad). GAPDH was used as loading control. For total protein normalization, labeling of the samples was initiated by carbocyanine (Cy)5 dye reagent (GE Healthcare) that had been pre-diluted (1:10) in ultrapure water. Samples were mixed and incubated at 21–24 °C for 5 min. The labeling reaction was terminated by adding Amersham WB loading buffer (GE Healthcare; 20 µl/sample) containing 40 µM dithiothreitol (Sigma, #10197777001). Equal amounts of the samples (20 µg/40 µl) were boiled at 95 °C for 3 min, and then loaded onto an Amersham WB gel card (13.5%). Electrophoresis (600 V, 42 min) and protein transfer onto PVDF membranes (100 V, 30 min) were at default settings in an integrated Amersham WB system (GE Healthcare). After blocking, membranes were incubated with primary antibodies overnight (Supplementary Table 3). Antibody binding was detected by species-specific Cy3-labeled secondary antibodies (1:1,000; GE Healthcare). Membranes were dried before scanning at 560 nm (Cy3) and 630 nm (Cy5) excitation. Automated image analysis was performed with the Amersham WB evaluation software with linear image optimization, if necessary.

### Prenatal exposure to an experimental diet

Female C57Bl6/J animals at 6 weeks of age were randomly assigned to either a hypercaloric diet enriched ~70-fold in ω-3 PUFAs or a standard diet. Prospective dams consumed the experimental diet for 8 weeks prior to conception. Diets were from Special Diet Services (United Kingdom) and quality controlled by mass-spectrometry (Supplementary Table 1). The effect of maternal nutrition on fetal brain development was evaluated at E18.5[35]. Embryos were collected from $n \geq 5$ different pregnancies to keep the number of independent observations sufficient for statistical analyses, sexed if appropriate, and used for histochemistry, proteomics, and molecular biology. The use of specific or mixed genders for particular experiments was specified throughout. Embryos were collected by Cesarean sections from mothers terminally anesthetized by isoflurane (5%, 1 L/min flow rate), weighed, and decapitated with their heads either immersed in 4% PFA

in 0.1 M PB (pH 7.4) overnight or snap-frozen in liquid $N_2$ and stored at −80 °C[52,65].

### Click-iT EdU labeling

Dams were intraperitoneally injected ("pulsed") with 5-ethynyl-2′-deoxyuridine (EdU, 33 mg/kg) at E14.5. At E18.5, $n = 3$ male embryos/group were selected by PCR genotyping (see ref.[24] for primer pairs and protocol). Their brains were immersion fixed in 4% PFA in PB overnight, rinsed in PB, and cryoprotected in 30% sucrose (in PB) for 48 h. Serial coronal sections (20 µm) were cut on a ThermoFisher NX70 cryostat, thaw-mounted onto fluorescence-free SuperFrost$^+$ glass slides, and stored at −20 °C until processing. EdU was visualized with Alexa 488-azide using the Click-iT labeling method (Life Technologies)[66].

### iTRAQ proteomics

Single cortical hemispheres ($n = 6$/group, mixed and balanced for sex from independent pregnancies) were obtained from E18.5 fetuses. Contralateral hemispheres were snap frozen for target validation by real-time PCR and/or Western blotting. Proteins were extracted by homogenization in lysis buffer (pH 10.0) of triethylammonium bicarbonate (25 mM, Sigma), $Na_2CO_3$ (20 mM, Sigma) in the presence of protease inhibitors (Complete™, EDTA-free Protease Inhibitor Cocktail; Sigma). Protein concentrations were determined by the bicinchoninic acid assay (Thermo Fisher). After quantification, proteins were precipitated in 6 volumes of ice-cold acetone. One hundred µg of proteins per sample were used for isobaric tagging for relative and absolute quantitation (iTRAQ) in an 8-plex layout per the manufacturer's instructions (ABSciex). In brief, proteins were denatured, reduced, alkylated, trypsin digested (Promega), individually labeled with appropriate iTRAQ tags, pooled, concentrated, re-suspended in 1.4 ml loading buffer (10 mM KH2PO4, pH 3.0 in 25% acetonitrile) and sonicated.

Peptides were separated by cation exchange chromatography on a PolySulfoethyl A column (PolyLC) over 30 min with a KCl gradient increasing up to 0.5 M, with 0.5 ml fractions collected. Fifteen fractions across the elution profile of similar peptide concentration were generated and concentrated (SpeedVac). Fractions were re-suspended in 0.1% trifluoroacetic acid (TFA) and desalted on C18 spin columns (PepClean C18, Thermo Scientific). Half of each fraction was then injected on an Acclaim PepMap 100 C18 trap and an Acclaim PepMap RSLC C18 column (Thermo Scientific), using a nanoLC Ultra 2D plus loading pump and nanoLC as-2 autosampler (Eksigent). The peptides were loaded onto the trap in a mixture of 98% water, 2% ACN and 0.05% TFA, washed for 20 min to waste before switching in line with the column, and were then eluted with a gradient of increasing ACN, containing 0.1% formic acid (2–20% ACN in 90 min, 20–40% in a further 30 min, followed by 98% ACN to clean the column before re-equilibration to 2% ACN). The eluate was sprayed into a TripleTOF 5600 electrospray tandem mass spectrometer (ABSciex) and analyzed in Information Dependent Acquisition (IDA) mode, performing 120 ms of MS followed by 80 ms MSMS analyses on the 20 most intense peaks seen by MS, with the "*Adjust collision energy when using iTRAQ reagent*" box ticked. The data files were processed by ProteinPilot 4.5 (Sciex) using the Paragon algorithm, searching against the SwissProt database (March 2013 edition). The following settings were selected: *sample type*: iTRAQ 8-plex (peptide labelled), *cysteine alkylation*: MMTS, *digestion*: trypsin, *instrument*: TripleTOF 5600, *species*: mouse, *ID focus*: biological modifications and amino acid substitutions, and *search effort*: thorough. Results were statistical evaluated in SPSS v. 21.

### Behavioral tests

Anxiety-like behavior in first-generation offspring was assessed in the elevated plus maze on P75 ($n = 6$ males/group from independent pregnancies) as described[67–70]. Mice were placed in the center of a

standard elevated plus maze apparatus (arms: 10 cm × 50 cm each) facing an open arm and allowed to explore the maze for 5 min. The distance travelled (cm) and the number of arm entries (*n*) were determined.

**Statistical analysis and figure design**

The number of independent samples is indicated in either the individual figure panels or their associated legends. Likewise, the number of animals is given where relevant. All values represent the means ± s.d. of independent experiments. Biological variation among the samples was similar throughout. Statistical significance was determined by either Student's *t*-test (for independent groups, two-tailed) or one-way and two-way analysis of variance (ANOVA) followed by either Bonferroni's or Tukey's *post-hoc* correction (for multiple groups). Statistical analysis was performed in Prism 8.0 (GraphPad). Statistical significance was defined throughout unless stated in bulk as follows: $*p < 0.05$, $**p < 0.01$, $***p < 0.001$, $****p < 0.0001$. Multi-panel images were assembled in CorelDraw 2022 using linear color, brightness, and contrast enhancement to increase visual clarity, if necessary.

# Data availability

All data relevant to judge and interpret this study were included in the paper and its accompanying Supplementary Information and Source Data Files. The mass spectrometry (proteomics) data have been deposited to ProteomeXchange (accession number: PDX066923), and are available at: https://www.ebi.ac.uk/pride/archive/projects/PXD066923 or for FTP download at: https://ftp.pride.ebi.ac.uk/pride/data/archive/2025/09/PXD066923. Single-cell RNA-seq datasets used in the current study are available in Gene Expression Omnibus (GEO SuperSeries GSE153164; https://www.ncbi.nlm.nih.gov/geo/query/acc.cgi?acc=GSE153164). Processed single-cell RNA-seq data are available at: https://harkany-lab.github.io/Cinquina_2024/cortex_visualisation.html. Source data are provided with this paper.

# Code availability

The code developed and used in this report has been published at https://harkany-lab.github.io/Cinquina_2024. A whole system setup was also made available for download. Data analysis using Bootstrap-coupled estimation were also deposited (https://harkany-lab.github.io/Cinquina_2024/methods.html).

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

## Acknowledgements

We thank J. Mulder and M. Uhlén (Karolinska Institutet) for anti-S100A6 and anti-CaCyBp antibodies, G.G. Kovacs (Medical University of Vienna) for human tissues, and S.L. Shirran, C.H. Botting (University of St. Andrews, United Kingdom; supported by the Wellcome Trust [BB/T017686/1]), M.A. Fuszard (Martin-Luther University, Germany), P. Poullet (Institut Curie & INSERM U900, France) for iTRAQ proteomics and analysis; and A. Fabisikova (Mass Spectrometry Center at the University of Vienna) for LC-MS/MS analysis. This work was supported by the Austrian Science Fund (P 34121-B; E.K.; 10.55776/COE16; T.H.), the National Brain Research Program of Hungary (2017-1.2.1-NKP-2017-00002, NAP2022-I-1/2022; A.A.), the Excellence Program for Higher Education of Hungary (TKP-EGA-25; A.A.), the Swedish Research Council (2023-03058; T.H.), Novo Nordisk Foundation (NNF23OC0084476; T.H.); Hjärnfonden (FO2022-0300; T.H.), European Research Council (FOODFORLIFE, 2020-AdG-101021016; T.H.), and an intramural research grant of the Medical Neuroscience Cluster of the Medical University of Vienna (2021-1; T.H.).

## Author contributions

T.H. conceived the project. V.C., V.D.M., A.V., and T.H. designed experiments. A.A., E.K., and T.H. procured funding. V.C., D.C., F.P., and E.K. performed in vivo and in vitro experiments and analyzed data. P.K. performed mass-spectrometry-based ligand identification. E.O.T. performed and supervised single-cell data analysis. V.C. and T.H. wrote the manuscript with input from all co-authors. Correspondence and requests for materials should be addressed to T.H. (tibor.harkany@ki.se or tibor.harkany@meduniwien.ac.at).

## Funding

## Competing interests

The authors declare no competing interests.
