## [Transparent Peer Review file · Nature Communications]

Astrocytes modulate neuronal development by S100A6 signaling

Corresponding Author: Professor Tibor Harkany

Version 0:

Reviewer comments:

Reviewer #1

(Remarks to the Author)

This manuscript by Cinquina et al. reports that cortical astrocytes release S100A6, a calcium-binding protein also known as 'calcyclin,' and inhibit neuritogenesis by binding to CaCyBp, a calcyclin binding protein expressed by cortical neurons. By reanalyzing single-cell RNA-seq data from the fetal somatosensory cortex, which was previously studied by Bella et al. in Nature (2021), the authors show that the expression pattern of S100A6 follows the time point of astrogliogenesis and is primarily expressed by glial cell types. Simultaneously, the authors suggest that CaCyBp is expressed in the neuronal lineage from the early embryonic period, by performing FISH, IHC, and Western blotting. The authors also suggest that S100A6 expression can be regulated in an activity-dependent manner using in vitro cultured primary astrocytes. Additionally, exposing recombinant S100A6 to in vitro primary neurons reduce neurite complexity, coinciding with a reduction in CaCyBp protein levels. Knocking down CaCyBp through AAV-mediated shRNA transfection inhibits neuritogenesis. The authors relate the limited neuronal morphogenesis to protein turnover at the endoplasmic reticulum (ER), as CaCyBp appears to localize in proximity to the ER. Knocking down CaCyBp reduces total protein content, including proteins involved in the ER stress pathway. Finally, they examined the role of environmental factors, especially eicosapentaenoic acid (EPA), in affecting cortical neuronal architecture by exposing pregnant mice to an EPA-enriched diet and comparing S100A6 and CaCyBp expression levels. During the developmental period, it is well known that astrocytes affect neuronal differentiation and synaptogenesis. The potential roles of novel astrocyte-derived factors that control neuritogenesis are of high interest. However, the data supporting a direct interaction between astrocytic S100A6 and neuronal CaCyBp are insufficient in the current version of the manuscript. The expression data are not conclusive, with no control data. Furthermore, the raw in vitro images/data were not presented, making it difficult to judge the authors' claims. Overall, this manuscript raises interesting questions but is not suitable for publication in Nature Communications.

Major Concerns:

1. The scRNA-seq data show that S100A6 is expressed in many different cell types during late embryonic days, and the authors noted its expression in astrocytes, oligodendrocytes, and glial progenitor cells. However, the expression of S100A6 should be better characterized. For example, is it oligodendrocytes or their precursor cells (OPCs) that express S100A6? Is S100A6 expressed only in fibrous astrocytes or in protoplasmic astrocytes as well? The in-situ data showing S100A6 in GFAP-expressing cells near the ventricular walls (in Fig. 1C) do not definitively demonstrate its expression in astrocytes since those cells could be radial glial cells. Moreover, many cells do not seem to co-express S100A6 and GFAP. Supplementary Figure 1's western raw data shows that S100A6 is almost absent in embryonic stages, unlike the graph.
2. In addition, in Fig. 1e, S100A6 appears to be expressed only in subsets of GFAP-expressing cells in the adult brain. Again, based on the examples of the cell images in Fig. 1e, it is difficult to judge whether S100A6 is really expressed in mature astrocytes, and if so, how heterogeneous the expression is in the astrocyte population.
3. Similar problems appear in CaCyBp expression as well. Control experiments showing its specific expression in neuronal cells should be presented for Fig. 1c, 1d, 1f, and 1g.
4. In Fig. 1h, it is not clear how the authors purified astrocytes vs. neurons. The control data is missing. This would be necessary for proving purity/enrichments after cell type-specific purification steps.
5. Many remarks were made based on inconclusive data, yet the tone of the conclusion seems very strong. For example, "Thus, we reveal the mutually exclusive expression of S100A6 and CaCyBp in astrocytes vs. neurons during corticogenesis and suggest that intercellular S100A6-to-CaCyBp signaling could constitute a novel mode of intercellular communication with S100A6 acting as a bona fide gliotransmitter." This should be rewritten to reflect the exact data, or the authors should provide more sufficient data to support such claims.

6. The increase of S100A6 secretion from astrocytes after glutamate treatment should be better examined. This conclusion is solely based on Western blot data which does not appear to be convincing. Would expressing S100A6-His tag in astrocytes and measuring His-tagged protein after glutamate treatment produce the same results? Additional experiments supporting the hypothesis that S100A6 can function as a gliotransmitter, should be presented.
7. The binding of S100A6 with neuronal CaCyBp is very poorly shown. Any recombinant protein added into cultured cells would be endocytosed to some degree. Thus, Fig. 2c does not really support the authors' claim. A more sophisticated binding assay should be performed.
8. The authors did not show any raw images of neurite complexity, making it impossible to judge the authors' claim. In fact, in Fig. S2c, neurons in this image do not seem to have reduced neurites after viral shCaCybp transfection.
9. shRNA for CaCybp shows just a 20% reduction in CaCybp expression, which is a very suboptimal knockdown efficiency for shRNA.
10. It is hard to visualize ER structure in Fig. 3a. Thus, it is difficult to conclude that CaCybp is preferentially localized at the ER.
11. It is not clear that KD of CaCyBp has any specific effects since all protein levels seem to be affected. Moreover, no direct data were presented about protein turnover.

Reviewer #2

(Remarks to the Author)

In this manuscript titled "Astrocytes modulate neuronal development by S100A6 signaling" the authors propose a new astrocyte-neuron communication pathway involved in controlling proteostasis and, as a consequence, the inhibition of neurogenesis. To demonstrate the existence of this pathway, the authors use available scRNA-seq data, and a series of immunostainings and molecular biology techniques to validate the expression and potential interactions of S100a6 (astrocytes) and CaCyBp (neurons) in both tissue samples and cultured cells. They also propose this pathway can be modulated in fetuses by the diet taken during pregnancy and that the changes in neurogenesis is due to alterations in proteostasis.

The discovery of astrocyte molecules that block neuronal maturation would be a new important addition to the field. This manuscript, suggest one potential pathway and mechanism. However, I think the data shown is mostly correlative and not enough to demonstrate the author's conclusions.

Major comments

- 1- When do the authors think this pathway would be important for neuronal development? The timepoints analysed are not consistent throughout the manuscript. E.g. Figure 1a,b focus of E10.5-P4, while the gene/protein expression validation is performed a P120 (Figure 1e,f and S1). Importantly, the mouse model experiments (Figure 3) are performed with E18.5 foetal brains, at this timepoint there are very few astrocytes in the brain, they are just starting to form. Why did the authors choose this timepoint if they would like to demonstrate that this is an astrocyte-neuron interaction? Would not be more appropriate to test it postnatally?
- 2- Experiments in Figure 1c-g will need quantification. The current images do not support the conclusions drawn by the authors. E.g. I cannot see S100a6 in Figure 1d. Also Figure 1e seems to show expression of S100a6 in GFAP-negative cells. Are they astrocytes or something else? What brain region is this? It is unclear if S100a6 and CaCyBp are expressed close to each other in the same brain region and at the same time. Co-staining of both proteins will need to be done to be able to conclude this is the case.
- 3- Line 100 -101 and Figure 2a, the authors state there is lack of cell death with glutamate treatment. However, the graph suggests otherwise. Could the authors please include the statistics for this analysis? Could they also indicate the number of biological replicates for This graph and Figure 2c please?
- 4- Figure 2, the addition of the recombinant S100a6-His tag protein to neuronal cultures is an interesting one, but in order to draw any conclusions it will need a control, e.g. addition of another recombinant protein. Moreover, the fact that the recombinant S100a6 could enter neurons doesn't demonstrate that astrocytes provide this protein to control CaCyBp expression and function. The authors could apply astrocyte conditioned media with or without S100a6 (removed or blocked with antibodies) and test the effects on CaCyBp and neurogenesis. Also, the authors could co-culture neurons with astrocytes that express a S100a6 shRNA or scramble. More direct evidence is needed to demonstrate this is an astrocyte-neuron communication pathway.
- 5- The authors conclude "our results describe a novel molecular axis of astrocyte-neuron communication, which is unique in limiting neuronal morphogenesis through the ER/UPS pathway". More conclusive experiments are needed to demonstrate the astrocyte S100a6 – neuron CaCyBp exists and that there is a direct link between this and specific proteostatic mechanisms. At the moment all conclusions are based on correlative data. A combination of knock down/out experiments for the specific molecules involved will need to be performed in a cell specific manner to better understand the relationship between the molecular processes investigated in this study.

Minor comments

- 1- Line 63: It would be helpful to mention on the main text that the scRNAseq comes from mouse.
- 2- Line 65-66, Re Figure 1a-b. Could the authors better justify why they think S100a6 is the most interesting candidate? There are several S100 genes that show a similar pattern of expression throughout development.
- 3- Figure 1h: Are these graphs normalized so the average of the first column is 1? If so, this should be mentioned in the figure legend.
- 4- Line 127: I believe Fig. 2b is called wrongly in this line.
- 5- Line 141: The authors state "without affecting either astrocyte numbers or viability (Supplementary Fig. 3e,f)". This is

incorrect, there is an increase in viability with EPA treatment.

6- Figure 3c, have the authors performed power calculations? There is a lot of variability in these graphs and it looks like an $n > 3$ will be needed to conclude that the non-differentially expressed genes, are indeed not differentially expressed.

7- Figure 3 states groups as standard or experimental. This is a bit unusual it would be more helpful to use tags that are more descriptive of the experiment, e.g. diet vs no diet.

8- Could the authors please clarify how they calculated total protein with only one band from the western blot?

Reviewer #3

(Remarks to the Author)

This is an interesting paper identifying a novel mechanism of astrocyte-neuron communication that regulates neuronal morphogenesis. The authors show that astrocytes express and release the protein S100A6 which acts on neuronally expressed calcyclin-binding protein (CaCyBp) to mediate changes in neuronal morphology through limiting neurite formation. Further, authors show that CaCyBp is present near ER sites and regulates unfolded protein response in neurons. S100A6 acts to inhibit CaCyBp, slowing protein turnover and thus generation of neurites. Additionally, CaCyBp levels are reduced, while S100A6 levels are increased in embryos exposed to eicosapentaenoic acid (EPA) in utero through maternal nutrition. The findings presented in this manuscript describe a very interesting astrocyte-neuron communication mechanism, linking astrocytes to neuronal morphogenesis. Overall, the manuscript is well written, and the data quality is good for the most part. However, some key points to strengthen the study's conclusions and interpretation are missing and should be addressed:

1. Related to data shown in Figure 1a, why does the study focus specifically on S100A6? While there is an increase in its expression at P4 compared to embryonic development, there are multiple additional S100 proteins that show the same developmental expression profile (such as S100A1, S100A4, S100A14 and others). Are they all expected to act via similar mechanism?

2. It is unclear at which age of animal development does the S100A6-CaCyBp interaction regulates neuronal morphogenesis. The data in Figure 1a show increased expression at P4 compared to embryonic ages, however, subsequent validation experiments shown in Figure 1e-f and suppl. Figure 1 show embryonic ages and adult (P120). It is unclear why the adult age was chosen for these experiments, since it is likely that this pathway is active during early postnatal development. It is important to validate cell type specific expression of these proteins at the appropriate age, such as P4 to support the RNAseq data, and the author's hypothesis. Since *gfap* is not a good marker of cortical astrocytes, this analysis should be done using *Aldh111*, *S100b* or *slc1a3* to label astrocytes during postnatal stages.

3. The images in Figure 1 c, d are unclear, and hard to see cell type-specific expression. It would be better to focus on one brain region (e.g., cortex) in this figure and show single channel images as well as the merged images to indicate cell type specific expression of these genes at both embryonic and later ages (such as P4).

4. Please provide quantification for images of data shown in Figure 1c, d, e, f.

5. In experiment shown in Figure 2c, authors state that S100A6 can also be visualized inside the cells (line 104 in the text). Can you please elaborate on how that conclusion was reached? It is not clear from the image shown in the figure. How does S100A6 secreted from astrocytes interact with the cytosolic CaCyBp in neurons, which seems to be the main location for this protein?

6. Please provide representative images of neurons used for morphology analysis shown in Figure 2e and g.

7. The efficiency of shRNA knockdown as shown in suppl. Figure 2d is very low (less than 50% decrease in protein expression). Is that due to high cell mortality? The example images show decreased cell density in the shCaCyBp treated group (suppl. 2c), though it is not quantified. Could cell mortality due to shRNA knockdown also explain the lower protein content and UPR response proteins decrease shown in Figure 3b-c?

8. Authors show that pups exposed to EPA through maternal diet exhibited increased S100A6 levels, and decreased CaCyBp levels, as well as changes in total protein and tubulin acetylation, which led the authors to conclude that S100A6-CaCyBp signaling may alter cortical architecture during development. A key experiment to support this conclusion is testing whether neuronal morphology (e.g., neurite outgrowth) is altered in vivo in the cortex of these pups at postnatal developmental stages as compared to pups born to females who consumed regular diet.

Minor points:

1. In the abstract (line 31) authors state .."guidance factors to promote neuronal differentiation".. is the intent here to say morphogenesis or neurite differentiation since neuronal cell differentiation occurs before gliogenesis?

2. In figures 1a-b scale bars, please provide numerical values (such as log₂ cpm or fpkm) for the gene expression plots, as it is not possible to assess expression levels from min and max labels.

Version 1:

Reviewer comments:

Reviewer #1

(Remarks to the Author)

1. Again, S100a6 expression in astrocytes is poorly presented in the revised manuscript. GFAP mRNA expression in Fig 2b is hardly visible in the cortical area, which is not surprising since cortical astrocytes express very low levels of GFAP. It is unclear why the authors did not use additional markers for astrocytes, such as Aldh1l1, S100beta, APOE, or EAAT1/2, to show colocalization of S100a6 with astrocytes. Moreover, the question of potential heterogeneous expression of S100a6 in astrocytic sub-populations was not properly investigated. This is an important point to support the authors' claim that S100a6 functions as an astrocytic signaling molecule in controlling neuronal protein turnover during general developmental stages.
2. S100a6 appears to be expressed in the oligodendrocyte lineage based on scRNA-seq data, but its expression pattern and contribution are not investigated or discussed. Without this information, how can the authors conclude that S100a6 is an astrocyte-derived factor?
3. To claim that S100a6-HA localization in N2A cells is related to CaCyBp, the authors should knock down CaCyBp in N2A cells and report changes in S100a6-HA localization. Otherwise, the current EM and MS data cannot serve as evidence for the S100a6-CaCyBp interaction.
4. I cannot acknowledge the reduced neurite length shown in the images provided by the authors.
5. In Fig 3, the authors mentioned that CaCyBp-like immunoreactivity typically accumulated at subplasmalemmal positions in mitochondria. However, in Fig 4, they also mentioned that CaCyBp is mostly located in the endoplasmic reticulum. Which one is correct?
6. S100a6 treatment induces downregulation of CaCyBp, leading to a reduction in overall protein content. However, there is no strong data supporting that this general protein content reduction is actually due to UPR responses. Without functional interrogation of UPR responses, the S100a6-CaCyBp interaction and its role in protein homeostasis through UPR remain as a hypothesis.

Reviewer #2

(Remarks to the Author)

I congratulate the authors for submitted a much-improved manuscript. The data is now more robust and supports the majority of the conclusions. I thank the authors for addressing many of my comments.

My main concern was summarized in the following comment "More conclusive experiments are needed to demonstrate the astrocyte S100a6 – neuron CaCyBp exists and that there is a direct link between this and specific proteostatic mechanisms. At the moment all conclusions are based on correlative data. A combination of knock down/out experiments for the specific molecules involved will need to be performed in a cell specific manner to better understand the relationship between the molecular processes investigated in this study." Although the authors have made a strong effort to improve the manuscript, I am afraid this manuscript is still lacking conclusive experiments that demonstrate a direct link between the astrocyte upregulation of S100a6 and neuronal downregulation of CaCyBp and other phenotypes. The manuscript is titled "Astrocytes modulate neuronal development by S100A6 signaling". Thus, it is essential that the data supports this idea.

For example:

- Most of the in vivo experiments are done at E18.5, when there are few astrocytes and clear expression of S100a6 in other cell types (Fig. 1e-g).
- The in situ hybridization figures (Fig. 2 and Extended Data 2) show that RNA of S100a6 and CaCyBp are in the same brain area, which is great, but they don't allow to assess in what cell type they are expressed.
- The evidence in vitro is helpful. However, it is still lacking proof that the effects on neurons with astrocyte conditioned media comes from astrocyte-released S100a6. This is because, (1) glutamate and EPA are still in the media when collected so they could have a direct effect on the neurons, (2) many other molecules could be released by astrocytes in response to glutamate and EPA, (3) only by removing S100a6 from the media, before incubation or knocking down S100a6 in the astrocytes we would be able to conclude that S100a6 has an effect on neurons in vitro.
- Thank you for doing the S100a4 experiment. It would be good to include this in the manuscript. Also, it would be useful to know if the neuron complexity is altered with S100a4. This will definitely conclude that there is a specific relationship between S100a6 and the morphology of neurons.

An additional comment that emerged from the reviewed manuscript is that the authors say "EPA aids the formation of neuronal membranes, enhances membrane fluidity, and ensures the efficient transmission of neural signals. EPA also influences gene expression to facilitate the growth and differentiation of neurons, synaptogenesis, and myelination", and other similar statements. However, the experiments suggest EPA would enhance S100a6 release and as a consequence block neuronal branching. How are these two conclusions compatible? This should be discussed in the discussion.

Overall, I think the results have potential interest, but they don't support the title "Astrocytes modulate neuronal development by S100A6 signaling", nor many of the conclusions that establish a direct link between astrocyte S100a6 and neuronal phenotypes. Either new experiments are included to conclude that, or a new title would be needed with rephrasing throughout the text to mention that the data suggests an interaction between astrocytes and neurons and that further experiments will be needed to establish robust conclusions.

Reviewer #3

(Remarks to the Author)

The authors did a good job addressing some of the reviewers' comments by restructuring the manuscript text and figures and performing additional experiments. However, some points raised by this reviewer were not sufficiently addressed, raising concerns about the study's conclusions and interpretation. Points to be addressed are listed below:

1. In the rebuttal, the authors provide a good explanation for why S100A6 was chosen over other family members for further analysis (reviewer#3, Q2), however, the manuscript text states that it was chosen due to increased expression from E18 to P4 (lines 95-99). As previously mentioned, additional family members also show a sharp increase at P4. It would be helpful to edit the text to emphasize why this protein was selected for further analysis.
2. The RNAseq data shows that cacybp is expressed by multiple cell types, including astrocytes at both E18 and P4 (Figure 1h, i). Yet the authors consider it to be specific to neurons. The in vivo validation experiments shown in Figure 2b are not convincing. Gfap signal is completely invisible or not shown and it is hard to see which cells co-express the markers. Figure 2c – bottom panel, does not appear like there is any colocalization based on the image. The findings in cell culture are encouraging, however, the validation in vivo needs to be further explored. The figure also lacks quantification (for example, how many map2 positive cells are colocalized with CaCyBp signal, or area or colocalized signal between 2 markers) and should be added.
3. The question whether S100A6 and CACYBP are expressed by astrocytes in vivo is insufficiently addressed. As mentioned in reviewer comments, gfap is not a good astrocyte marker for the cortex. The authors mention in the rebuttal that experiments using S100b and Aldh11 were performed in response to reviewer's comments – however, I could not find these in the revised manuscript. An experiment, similar to that shown in Figure 2c and extended data Figure 2c, d needs to be performed with S100b or Aldh11 as an astrocytic marker (similarly to what was done using neun as neuronal marker). The experiments in extended data Figure 2g are showing that these markers are expressed by astrocyte in culture, however, it does not prove the astrocytic origin of S100A6 or CACYBP in vivo.
4. Extended data Figure 2e – the zoomed-in region from box labeled 1 is not showing the same area of the image.
5. For all images in Figure 2 and extended data Figure 2 – it would be helpful to add arrows to the images to guide the reader to the relevant cell type / signal that is discussed in the text or figure legend.
6. The images of neurons provided in Figure 3h (in response to reviewer's comments Q7) are only shown for an experiment in which cacybp is downregulated. The images shown (Figure 3h) are hard to interpret as only the length of branches is analyzed. However, it looks from the image that the neuron treated with cacybp shRNA shows increased branch number. As the implications of S100A6-CACYBP signaling on neuronal morphogenesis is one of the key findings of this study, performing a more in-depth analysis and providing stronger evidence for morphological changes upon these treatments is very important. Authors should provide images of neurons treated with S100A6 (Figure 3f) so that they can be compared with the morphological differences shown upon the knockdown of cacybp. Sholl analysis to determine neuronal morphological complexity would be beneficial for these experiments.
7. The downregulation in map2 and gap43 signals shown in extended data Figure 6 is quite dramatic. Do these mice exhibit severe brain developmental abnormalities after birth?
8. The authors state that the number and positioning of the neurons is altered upon the experimental diet, based on downregulated expression of map2 and gap43 (lines 307-308). It is unclear how this conclusion was reached based on the data, since the number of dapi+ cells is not altered. A detailed analysis of neuronal numbers and positions within the cortex should be conducted, or the text should be revised to more accurately describe the experimental finding.

Version 2:

Reviewer comments:

Reviewer #1

(Remarks to the Author)

Although the authors have made efforts to address the previous concerns, I remain uncertain about some of the conclusions presented in the manuscript.

The authors emphasize that S100A6 could serve as an astrocyte-specific gliotransmitter. However, based on their scRNA-seq and fluorescent in situ hybridization data, it is evident that S100A6 is not exclusively expressed in astrocytes, but also in other cell types. Given that S100A6 can be secreted by various cells during both embryonic and postnatal development, how can the authors claim that S100A6 functions as an astrocyte-specific gliotransmitter?

In addition, the conclusions drawn from the proteostasis-related experiments and eicosapentaenoic acid (EPA) treatment appear to be largely correlative, lacking direct causal evidence. Most of the proteins examined seem to be downregulated following S100A6 treatment or Cacybp knockdown. How, then, can the authors attribute these effects specifically to the unfolded protein response (UPR) or ubiquitin-proteasome system (UPS)?

Finally, although the levels of S100A6 and Cacybp were altered by EPA treatment, this alone does not support the conclusion that the observed outcomes—particularly in the in vivo experiments—are mediated through the S100A6–Cacybp signaling pathway.

Reviewer #2

(Remarks to the Author)

I thank the authors for carefully considering my comments and the effort they have put in addressing them. I think the manuscript has greatly improved and I would be happy for it to be published. Thanks!

Reviewer #3

(Remarks to the Author)

The authors have adequately addressed all my comments. I support the acceptance of this manuscript for publication.

RE: Point-by-point replies to the Reviewers' comments

Reviewer #1:

General: We appreciated your recognition of the importance of our study, including its focus and results. We thank your many helpful and critical comments that helped us to improve our narrative and analyses. Please find our point-by-point answers to your queries below.

Q1: 'This manuscript by Cinquina et al. reports that cortical astrocytes release S100A6, a calcium-binding protein also known as 'calcyclin,' and inhibit neuritogenesis by binding to CaCyBp, a calyculin binding protein expressed by cortical neurons. By reanalyzing single-cell RNA-seq data from the fetal somatosensory cortex, which was previously studied by Bella et al. in Nature (2021), the authors show that the expression pattern of S100A6 follows the time point of astroglialogenesis and is primarily expressed by glial cell types. Simultaneously, the authors suggest that CaCyBp is expressed in the neuronal lineage from the early embryonic period, by performing FISH, IHC, and Western blotting. The authors also suggest that S100A6 expression can be regulated in an activity-dependent manner using in vitro cultured primary astrocytes. Additionally, exposing recombinant S100A6 to in vitro primary neurons reduce neurite complexity, coinciding with a reduction in CaCyBp protein levels. Knocking down CaCyBp through AAV-mediated shRNA transfection inhibits neuritogenesis. The authors relate the limited neuronal morphogenesis to protein turnover at the endoplasmic reticulum (ER), as CaCyBp appears to localize in proximity to the ER. Knocking down CaCyBp reduces total protein content, including proteins involved in the ER stress pathway. Finally, they examined the role of environmental factors, especially eicosapentaenoic acid (EPA), in affecting cortical neuronal architecture by exposing pregnant mice to an EPA-enriched diet and comparing S100A6 and CaCyBp expression levels. During the developmental period, it is well known that astrocytes affect neuronal differentiation and synaptogenesis. The potential roles of novel astrocyte-derived factors that control neuritogenesis are of high interest. However, the data supporting a direct interaction between astrocytic S100A6 and neuronal CaCyBp are insufficient in the current version of the manuscript. The expression data are not conclusive, with no control data. Furthermore, the raw in vitro images/data were not presented, making it difficult to judge the authors' claims. Overall, this manuscript raises interesting questions but is not suitable for publication in Nature Communications.'

We appreciated all the observations that you have deemed were important to come to your conclusions. With regard to your specific critique, *i)* we have significantly strengthened the hypothesis that a direct interaction between S100A6-CaCyBp exists by means of *electron microscopy* and *mass-spectrometry*; *ii)* performed *in situ hybridization* and *immunohistochemistry* for age matching, cell identification, context dependence, and provided additional controls, and *iii)* inserted raw images on cell morphology. Given the extent of our revisions, including experimentation, we are confident to have addressed your queries appropriately, allowing you to support publication of our report in the journal.

Q2: 'Major Concerns: 1. The scRNA-seq data show that S100A6 is expressed in many different cell types during late embryonic days, and the authors noted its expression in astrocytes, oligodendrocytes, and glial progenitor cells. However, the expression of S100A6 should be better characterized. For example, is it oligodendrocytes or their precursor cells (OPCs) that express S100A6? Is S100A6 expressed only in fibrous astrocytes or in protoplasmic astrocytes as well? The in-situ data showing S100A6 in GFAP-expressing cells near the ventricular walls (in Fig. 1C) do not definitively demonstrate its expression in astrocytes since those cells could be radial glial cells. Moreover, many cells do not seem to co-express S100A6 and GFAP. Supplementary Figure 1's western raw data shows that S100A6 is almost absent in embryonic stages, unlike the graph.'

j) We appreciated your request to improve our analysis. We have done so: instead of relying on an open-label data editor from the group that had provided the single-cell RNA-seq data on cortical development, we have downloaded the raw data and re-analyzed those by using the Seurat pipeline (see also from our group: Romanov et al., Nature (2020); Benevento et al., Nature (2024), Romanov et al., Nat Neurosci (2017) and others). Thereby, we have significantly improved cell identification, clustering, and the distinction amongst cellular subtypes. The new level of analysis allowed us to focus on astrocytes that were identified through a set of stable genes (*Aldh1l1*, *Apoe*; *Aqp4*; **Extended Data Fig. 1a,b**; see also: Tretiakov et al., Research-square (2024) for astrocyte identities).

Figure 1. GFAP⁺ astrocyte morphologies and their co-expression of S100A6 in adult brain.

ii) In terms of fibrous vs. protoplasmic astrocytes, these are anatomical/cellular definitions that cannot be faithfully resolved by single-cell RNA-seq. Immunohistochemistry did not give a clear distinction for S100A6 expression *in vitro/in vivo* either (see **Figure 1 herein**). Moreover, since we do not have a general marker through which we could visualize every astrocyte, any detailed analysis would be inherently limited to subsets. Lastly, our study builds a mechanistic hypothesis on an intercellular interaction rather than being a quantitative neuroanatomy study, thus – in our view – negating the need for a detailed morphometric study.

iii) Only a subset of astrocytes express GFAP under physiological conditions. This is why S100 β is historically used as a more pleiotropic marker. In our hands, *Apoe* was even more superior (**Extended Data Fig. 1b**). Nevertheless, and for illustrations only, we have selected cells co-localizing GFAP (**Fig. 2b**), given GFAP is the most-accepted astrocyte marker in the scientific community. We would also emphasize that our study is in a physiological context. At no point have we used experimental paradigms that could have transformed astrocytes (which would be associated with their morphological change). Thus, comparisons based on arbitrary morphologies (that is, we have no prior knowledge as to why two cells were different) would be potentially misleading. *iv*) Please note that the graphs associated with the Western blots provided **normalized values**, with E14.5 being “1” (that is, the internal standard for comparison). Therefore, the actual amount of a given protein can be near-zero at E14.5. Thus, the normalized graphs are correct.

Q3: ‘2. In addition, in Fig. 1e, S100A6 appears to be expressed only in subsets of GFAP-expressing cells in the adult brain. Again, based on the examples of the cell images in Fig. 1e, it is difficult to judge whether S100A6 is really expressed in mature astrocytes, and if so, how heterogeneous the expression is in the astrocyte population.’

Since the study focuses on calcyclin-binding protein (CaCyBp), we have initially placed less emphasis on localizing S100A6 in terms of gross neuroanatomy. In the revised manuscript, **Fig. 2b shows high-resolution *in situ* hybridization** for *S100a6*, and its overlap with *Gfap*. Moreover, **Extended Data Fig. 2a,b** show survey images for S100A6 and CaCyBp to allow for anatomical reconstructions of the cortical structures, and broader views of the cortical cell types in question (GFAP and NeuN were used as markers for astrocytes and neurons, respectively). The **contentious image using GFAP co-localization in adult brain was removed** because it was incompatible with the overall hypothesis and developmental biology rationale. Thus, we have precisely localized S100A6 in the fetal and neonatal mouse (and human) brain.

Q4: ‘3. Similar problems appear in CaCyBp expression as well. Control experiments showing its specific expression in neuronal cells should be presented for Fig. 1c, 1d, 1f, and 1g.’

Thank you for raising this point. We have **updated the figure** after performing a renewed round of immunohistochemistry (combined with CaCyBp, as well as NeuN, MAP2, and calretinin as neuronal markers). Our **revised manuscript contains data from both embryonic (E18.5) and postnatal (P4) mice**, as requested, to match the biochemical measurements. We have also revised **Extended Data Fig. 2**, which now includes survey images of the cerebral cortex, allowing judgement on cellular heterogeneity, abundance, and layer-specific distribution, if any. Such images were also included for humans. Thus, **Fig. 2** and **Extended Data Fig. 2** together provide detailed and precise information on CaCyBp expression in neurons.

Q5: '4. In Fig. 1h, it is not clear how the authors purified astrocytes vs. neurons. The control data is missing. This would be necessary for proving purity/enrichments after cell type-specific purification steps.'

We have updated the '**Methods**' section to describe cell type enrichment. In addition, control data on **astrocyte- vs. neuron-specific gene expression** after purification were included in **Extended Data Fig. 2g**.

Q6: '5. Many remarks were made based on inconclusive data, yet the tone of the conclusion seems very strong. For example, "Thus, we reveal the mutually exclusive expression of S100A6 and Ca-CyBp in astrocytes vs. neurons during corticogenesis and suggest that intercellular S100A6-to-Ca-CyBp signaling could constitute a novel mode of intercellular communication with S100A6 acting as a bona fide gliotransmitter." This should be rewritten to reflect the exact data, or the authors should provide more sufficient data to support such claims.'

Since this manuscript was transferred to Nature Communications, its original format disallowed more figures/lengthier text. In accord with your request, **we have strengthened the data experimentally rather than by toning down our statements**. Fortunately, our original hypotheses hold even after the extensive revisions we have made. Therefore, we are of the view that the data and narrative are cohesive as is.

Q7: '6. The increase of S100A6 secretion from astrocytes after glutamate treatment should be better examined. This conclusion is solely based on Western blot data which does not appear to be convincing. Would expressing S100A6-His tag in astrocytes and measuring His-tagged protein after glutamate treatment produce the same results? Additional experiments supporting the hypothesis that S100A6 can function as a gliotransmitter, should be presented.'

We have followed your advice and performed additional experiments: i) we have **extended the analysis to 24h (Fig. 3a)**, ii) performed **viability assays** by imaging (**Fig. 3b**) and biochemically (**Extended Data Fig. 3a**) to exclude cell death/damage. Indeed, we performed ELISAs on media collected from astrocytes after 6h and 24h post-treatment with glutamate to validate the release of S100A6, with an ~2-fold increase in S100A6 after 6h, and ~4-fold increase after 24h (**Fig. 3a**). We have also performed the same experiments upon eicosapentaenoic acid (**EPA**) stimulation (**Fig. 5a-c, Extended Data Fig. 5**) with three ligand concentrations. We have found a similar magnitude and dynamics of S100A6 release *in vitro*. In sum, these data support our initial hypothesis and original data. **Control data are in Extended Data Fig. 3b-f**. Thus, we are confident to have a sufficiently strong case for the release of S100A6.

Q8: '7. The binding of S100A6 with neuronal CaCyBp is very poorly shown. Any recombinant protein added into cultured cells would be endocytosed to some degree. Thus, Fig. 2c does not really support the authors' claim. A more sophisticated binding assay should be performed.'

Indeed, this point is central to our hypothesis. Therefore, we have performed i) new **super-resolution images** in Neuro-2a cells that express CaCyBp but not endogenous S100A6 **to show internalization (Fig. 3c)**, ii) **electron microscopy** on cultured cells to resolve the subcellular localization of His-tagged S100A6, which appeared particularly enriched around mitochondria (**Fig. 3d**), as well as **mass-spectrometry** on intracellular (cytosol) fractions after pulse-chase experiments using recombinant S100A6 (**Fig. 3e**).

Figure 2. mRNA levels for *Cacybp* (a) and ER stress markers (b) in E18.5 cortical neurons after exposure to S100A4 (1 μ g/ μ l) for 24h.

At the same time, we have used S100A4 to suggest that not all S100 proteins use the same signaling pathway. *i)* Unlike S100A6, recombinant S100A4 does not affect *Ca-cybp* mRNA expression *in vitro* (see **Figure 2a** herein). *ii)* Likewise, equimolar concentrations of S100A4 did not modify the expression of the molecular components of the UPS pathway either (see **Figure 2b** herein). Thus, we argue for selective signaling events triggered by S100A6 when available extracellularly.

Q9: '8. The authors did not show any raw images of neurite complexity, making it impossible to judge the authors' claim. In fact, in Fig. S2c, neurons in this image do not seem to have reduced neurites after viral shCaCybp transfection.'

Raw (illustrative) images were added to **Fig. 3h**.

Q10: '9. shRNA for CaCybp shows just a 20% reduction in CaCybp expression, which is a very suboptimal knockdown efficiency for shRNA.'

Indeed, our *in vitro* knock-down was incomplete. It is therefore the more encouraging that we have obtained meaningful results. This underscores the **significance of the findings** and highlights the robustness of our experimental approach. Notably, viral knock-down is rarely, if at all, near-complete when cell death usually becomes a complicating factor. Therefore, **we are glad to have titrated a system** in which efficacy vs. cell survival could be closely matched.

Q11: '10. It is hard to visualize ER structure in Fig. 3a. Thus, it is difficult to conclude that CaCybp is preferentially localized at the ER.'

Fig. 4b and **Extended Data Fig. 4a** were revised to show ER at high resolution, with semitransparent coloring used to pinpoint the ER. Moreover, we have performed cell fractionation, allowing Ca-CyBp localization to the ER fraction (**Fig. 4a**).

Q12: '11. It is not clear that KD of CaCyBp has any specific effects since all protein levels seem to be affected. Moreover, no direct data were presented about protein turnover.'

Figure 3. Western blot detection of USP9X and USP7 in the cortices of male embryos at E18.5 in control and after exposure to an experimental diet.

USP9X were tested in *in vitro* to emphasize the role of S100A6 in limiting protein turnover. We found a downregulation of USP9X (less so USP7; **Fig. 4e**) **knocking down CaCyBp**, which was interpreted as one of the possible causes of reduced total cellular protein contents. Downregulation of USP7 was also obtained **after S100A6 treatment (Extended Data Fig. 4d) and in cortical tissues of embryos after experimental diets to the maters (see Figure 3 herein)**. Cumulatively, these data reinforce our conclusion.

Impaired protein folding, the primary cause of ER stress, leads to increased protein degradation through ubiquitination. In contrast, deubiquitination contributes to protein quality control mechanisms by either rescuing ubiquitinated proteins from degradation or reversing the ubiquitination of misfolded or damaged proteins. By preventing the indiscriminate degradation of functional proteins, deubiquitinases maintain cellular proteostasis and prevent the accumulation of cytotoxic protein aggregates. USP7 and USP9X are abundant deubiquitinating enzymes. **Both USP7 and**

Reviewer #2:

General: Thank you for your supportive and constructive view on our manuscript. We have certainly appreciated the depth of your insightful review, and comments. Please find our succinct answers to both your major and minor queries as follows.

Q1: 'In this manuscript titled "Astrocytes modulate neuronal development by S100A6 signaling" the authors propose a new astrocyte-neuron communication pathway involved in controlling proteostasis and, as a consequence, the inhibition of neuritogenesis. To demonstrate the existence of this pathway, the authors use available scRNA-seq data, and a series of immunostainings and molecular biology techniques to validate the expression and potential interactions of S100a6 (astrocytes) and CaCyBp (neurons) in both tissue samples and cultured cells. They also propose this pathway can be modulated in foetuses by the diet taken during pregnancy and that the changes in neuritogenesis is due to alterations in proteostasis. The discovery of astrocyte molecules that block neuronal maturation would be a new important addition to the field. This manuscript, suggest one potential pathway and mechanism. However, I think the data shown is mostly correlative and not enough to demonstrate the author's conclusions.'

Thank you for your interest in our findings. We have performed many additional experiments to move our findings from 'correlative' to 'conclusive'. We hope you agree with us that we have achieved just that.

Q2: 'Major comments: 1- When do the authors think this pathway would be important for neuronal development? The timepoints analysed are not consistent throughout the manuscript. E.g. Figure 1a,b focus of E10.5-P4, while the gene/protein expression validation is performed a P120 (Figure 1e,f and S1). Importantly, the mouse model experiments (Figure 3) are performed with E18.5 foetal brains, at this timepoint there are very few astrocytes in the brain, they are just starting to form. Why did the authors choose this timepoint if they would like to demonstrate that this is an astrocyte-neuron interaction? Would not be more appropriate to test it postnatally?'

We think this pathway is physiologically important during neonatal synaptogenesis. Moreover, it could contribute to the structural plasticity of synapses (at more confined positions) also in adulthood. Yet the latter hypothesis needs thorough testing. *i)* **We have revised all figures to consistently focus on the time-points illustrated in Fig. 1.** We chose **E18.5** as the primary time-point for embryogenesis and **P4** for neonatal life. Our biochemical analyses concentrated on embryonic stages, during which both proteins are expressed. *ii)* During embryogenesis, protein turnover plays a crucial role in neuronal development, including neurogenesis, cell differentiation, morphogenesis, axon guidance and pathfinding, and synaptic plasticity. Therefore, our *in vivo* experiments were conducted at E18.5, allowing us to detect both CaCyBp and S100A6 by Western blotting, qPCR, and iTRAQ. *iii)* We have conducted single-cell RNA-seq and qPCR analyses to demonstrate the expression of astrocyte markers (e.g., *S100b*, *Aldh111*) at this time point.

Q3: '2- Experiments in Figure1c-g will need quantification. The current images do not support the conclusions drawn by the authors. E.g. I cannot see S100a6 in Figure 1d. Also Figure 1e seems to show expression of S100a6 in GFAP-negative cells. Are they astrocytes or something else? What brain region is this? It is unclear if S100a6 and CaCyBp are expressed close to each other in the same brain region and at the same time. Co-staining of both proteins will need to be done to be able to conclude this is the case.'

We have **revised the figures** to show the presence of both CaCyBp and S100A6 in the same cortical region both at E18.5 and P4. Moreover, we have **performed elaborate single-cell RNA-seq data analysis** to quantify RNA expression, and assign it to specific cellular lineages. These data are in **Fig. 1 and Extended Data Fig. 1.**

Q4: '3- Line 100 -101 and Figure 2a, the authors state there is lack of cell death with glutamate treatment. However, the graph suggests otherwise. Could the authors please include the statistics for this analysis? Could they also indicate the number of biological replicates for This graph and Figure 2c please?'

We have revised the figure, extended the treatment time, performed an endpoint MTT assay, and attached a table with statistics (source data file). The experiment was also repeated to increase the number of biological replicates. Thus, we are confident to have a tightly-controlled and precisely-executed experiment with reliable data.

Q5: '4- Figure 2, the addition of the recombinant S100a6-His tag protein to neuronal cultures is an interesting one, but in order to draw any conclusions it will need a control, e.g. addition of another recombinant protein. Moreover, the fact that the recombinant S100a6 could enter neurons doesn't demonstrate that astrocytes provide this protein to control CaCyBp expression and function. The authors could apply astrocyte conditioned media with or without S100a6 (removed or blocked with antibodies) and test the effects on CaCyBp and neuritogenesis. Also, the authors could co-culture neurons with astrocytes that express a S100a6 shRNA or scramble. More direct evidence is needed to demonstrate this is an astrocyte-neuron communication pathway.'

Thank you for addressing these pertinent concerns. *j*) We have used recombinant S100A4 as a control protein. Data are shown in **Fig. 2** (replies to Reviewer 1, page 4 herein). Given that **CaCyBp has been demonstrated to bind S100A6 but not S100A4**, we specifically targeted S100A4 to investigate its potential effect on ER stress. Our findings indicate that neither mRNA levels of CaCyBp nor ER stress markers were significantly modulated by S100A4. These data strengthen the specificity of our experiments using S100A6. *ii*) To further investigate astrocyte-neuron communication, we cultured cortical neurons and subsequently exposed them to astrocyte-conditioned media, with or without S100A6 pre-treatment (**Fig. 5d-h**). Initially, we conducted an ELISA test, revealing that astrocytes released S100A6 into the medium as early as 6h after glutamate (and also EPA) treatment. We then profiled the expression of ER markers when using conditioned media. Thus, we have arrived to stronger conclusions, as suggested.

Q6: '5- The authors conclude "our results describe a novel molecular axis of astrocyte-neuron communication, which is unique in limiting neuronal morphogenesis through the ER/UPS pathway". More conclusive experiments are needed to demonstrate the astrocyte S100a6 – neuron CaCyBp exists and that there is a direct link between this and specific proteostatic mechanisms. At the moment all conclusions are based on correlative data. A combination of knock down/out experiments for the specific molecules involved will need to be performed in a cell specific manner to better understand the relationship between the molecular processes investigated in this study.'

During the revision process, we included both **USP7 and USP9X** to highlight the role of S100A6 in regulating protein turnover. **Both USP7 and USP9X were tested in *in vitro*** to emphasize the role of S100A6 in limiting protein turnover. We found a downregulation of USP9X (less so USP7; **Fig. 4e**) **knocking down CaCyBp**, which was interpreted as one of the possible causes of reduced total cellular protein contents. Downregulation of USP7 was also obtained **after S100A6 treatment (Extended Data Fig. 4d) and in cortical tissues of embryos after experimental diets to the mothers**. Data are shown in Fig. 3 (replies to Reviewer 1, page 5 herein). Cumulatively, these data reinforce our conclusion.

Q7: 'Minor comments: 1- Line 63: It would be helpful to mention on the main text that the scRNAseq comes from mouse.'

The **text has been revised** as requested.

Q8: '2- Line 65-66, Re Figure 1a-b. Could the authors better justify why they think S100a6 is the most interesting candidate? There are several S100 genes that show a similar pattern of expression throughout development.'

While several S100 genes exhibit similar patterns of expression throughout development, we believe that S100A6 stands out as an intriguing candidate: *i)* S100A6 is involved in a broad range of cellular processes, including Ca²⁺ signaling, cytoskeletal dynamics, cell cycle progression, and cell differentiation; *ii)* its functions suggest a complex regulatory role that extends beyond typical S100 proteins; *iii)* S100A6 interacts with CaCyBp, which we find near-ubiquitously expressed in neurons. Understanding the mechanism of S100A6 and CaCyBp interaction can produce insights in astrocytes-neuron communication, relevant for circuit development. **These reasons were verbalized in the revised manuscript.**

Q9: '3- Figure 1h: Are these graphs normalized so the average of the first column is 1? If so, this should be mentioned in the figure legend.'

The figure **legend was updated**.

Q10: '4- Line 127: I believe Fig. 2b is called wrongly in this line.'

The **text has been revised**.

Q11: '5- Line 141: The authors state "without affecting either astrocyte numbers or viability (Supplementary Fig. 3e,f)". This is incorrect, there is an increase in viability with EPA treatment.'

The text has been revised to improve clarity. We have also **increased the number of biological replicates** to justify a lack of cell death after EPA treatment.

Q12: '6- Figure 3c, have the authors performed power calculations? There is a lot of variability in these graphs and it looks like an n>3 will be needed to conclude that the non-differentially expressed genes, are indeed not differentially expressed.'

We have **increased the number of biological replicates**, as requested.

Q13: '7- Figure 3 states groups as standard or experimental. This is a bit unusual it would be more helpful to use tags that are more descriptive of the experiment, e.g. diet vs no diet.'

We wish to keep these identifiers because we are comparing an **experimental diet** (with altered composition) with the **standard laboratory chow**. 'No diet' would not be entirely correct because the standard chow also contains the same components yet at other concentrations.

Q14: '8- Could the authors please clarify how they calculated total protein with only one band from the western blot?'

We apologize for the confusion regarding these results. Given the changes in the total protein content that we have detected by Western blotting in our *in vitro* and *in vivo* experiments, we have normalized the expression of target proteins to GAPDH. **Figures and text have been revised** to reflect the correct workflow.

Reviewer #3:

General: Thank you for your supportive comments on our manuscript, and for recognizing its strengths. We have carefully addressed your remarks and went at length to design and execute experiments if and when required. Please find our responses to your specific queries as follows.

Q1: 'This is an interesting paper identifying a novel mechanism of astrocyte-neuron communication that regulates neuronal morphogenesis. The authors show that astrocytes express and release the protein S100A6 which acts on neuronally expressed calyculin-binding protein (CaCyBp) to mediate changes in neuronal morphology through limiting neurite formation. Further, authors show that Ca-CyBp is present near ER sites and regulates unfolded protein response in neurons. S100A6 acts to inhibit CaCyBp, slowing protein turnover and thus generation of neurites. Additionally, CaCyBp levels are reduced, while S100A6 levels are increased in embryos exposed to eicosapentaenoic acid (EPA) in utero through maternal nutrition. The findings presented in this manuscript describe a very interesting astrocyte-neuron communication mechanism, linking astrocytes to neuronal morphogenesis. Overall, the manuscript is well written, and the data quality is good for the most part. However, some key points to strengthen the study's conclusions and interpretation are missing and should be addressed.'

We have appreciated your positive assessment. Our specific replies to your queries follow:

Q2: '1. Related to data shown in Figure 1a, why does the study focus specifically on S100A6? While there is an increase in its expression at P4 compared to embryonic development, there are multiple additional S100 proteins that show the same developmental expression profile (such as S100A1, S100A4, S100A14 and others). Are they all expected to act via similar mechanism?'

S100A6 is involved in a broad range of cellular processes, including cell cycle progression, cytoskeletal dynamics, and cell differentiation, thus being set apart from other S100 family members. What makes S100A6 particularly intriguing is its interaction with calyculin-binding protein (CaCyBp) and its downstream E3 ubiquitin ligases. Our study is particularly strong in noting the **near-ubiquitous expression of CaCyBp in developing cortical neurons in both mouse and human brain**. The **complementarity of S100A6 and CaCyBp in astrocytes and neurons**, respectively, makes a particular case for intercellular signaling. Not all S100 proteins are released. Moreover, and to the best of our knowledge, only S100A6 binds CaCyBp. Differences are also evident when comparing S100A4 (see **Fig. 2 at Reviewer 1**, p. 4), which neither binds CaCyBp nor affects the UPS machinery. Thus, we think there is due specificity in action amongst S100 proteins.

Q3: '2. It is unclear at which age of animal development does the S100A6-CaCyBp interaction regulates neuronal morphogenesis. The data in Figure 1a show increased expression at P4 compared to embryonic ages, however, subsequent validation experiments shown in Figure 1e-f and suppl. Figure 1 show embryonic ages and adult (P120). It is unclear why the adult age was chosen for these experiments, since it is likely that this pathway is active during early postnatal development. It is important to validate cell type specific expression of these proteins at the appropriate age, such as P4 to support the RNAseq data, and the author's hypothesis. Since *gfap* is not a good marker of cortical astrocytes, this analysis should be done using *Aldh111*, *S100b* or *slc1a3* to label astrocytes during postnatal stages.'

Thank you for raising this point, which we undoubtedly agree with. We **have updated all figures** to center our data on E18.5 and P4. Our biochemical analysis was primarily directed towards embryonic stages, where both proteins under investigation are already expressed. We have also performed additional experiments using *Aldh111* and *S100b* as postnatal markers for cortical astrocytes. Accordingly, **Fig. 2, Extended Data Fig. 1 and 2 were revised, and the text updated**.

Q4: '3. The images in Figure 1 c, d are unclear, and hard to see cell type-specific expression. It would be better to focus on one brain region (e.g., cortex) in this figure and show single channel images as well as the merged images to indicate cell type specific expression of these genes at both embryonic and later ages (such as P4).'

This has been done as requested. **Overview/single channel images were placed in Extended Data Fig. 2.**

Q5: '4. Please provide quantification for images of data shown in Figure 1c, d, e, f.'

We have not done morphometric quantification, which would not have provided reliable single-cell resolution. Instead, **we have re-processed large-scale single-cell RNA-seq data** through which we could resolve expression levels of many astrocytes, with data presented in **Fig. 1** and **Extended Fig. 1**, using many more marker genes than would have been conceivable through immunohistochemistry.

Q6: '5. In experiment shown in Figure 2c, authors state that S100A6 can also be visualized inside the cells (line 104 in the text). Can you please elaborate on how that conclusion was reached? It is not clear from the image shown in the figure. How does S100A6 secreted from astrocytes interact with the cytosolic CaCyBp in neurons, which seems to be the main location for this protein?'

Indeed, the original images might have been perceived as sub-optimal. During the revision, we have performed **super-resolution microscopy, electron microscopy, and mass-spectrometry (Fig. 3c-e; Extended Data Fig. 3d-f)** to justify our conclusions. These data show the accumulation of recombinant S100A6 intracellularly at specific subcellular sites, which are relevant to the intracellular signaling mechanisms we have outlined. For the specific interaction, we believe that CaCyBp might be inserted also in the cell membrane or internalized S100A6 could bind to intracellular CaCyBp.

Q7: '6. Please provide representative images of neurons used for morphology analysis shown in Figure 2e and g.'

The requested images are **part of revised Fig. 3h.**

Q8: '7. The efficiency of shRNA knockdown as shown in suppl. Figure 2d is very low (less than 50% decrease in protein expression). Is that due to high cell mortality? The example images show decreased cell density in the shCacybp treated group (suppl. 2c), though it is not quantified. Could cell mortality due to shRNA knockdown also explain the lower protein content and UPR response proteins decrease shown in Figure 3b-c?'

Thank you for raising this point. Although the decrease in CaCyBp expression was indeed moderate, it is heartening to see that our study still yielded meaningful results. This reaffirms the importance of our findings and validates the strength of our experimental approach. The MTT assay did not show any variation in cell viability, as compared to control. This is in line with the increase of acetylate-tubulin. Thus, we concluded that cell death is unlikely a biasing factor in these experiments.

Q9: '8. Authors show that pups exposed to EPA through maternal diet exhibited increased S100A6 levels, and decreased CaCyBp levels, as well as changes in total protein and tubulin acetylation, which led the authors to conclude that S100A6-CaCyBp signaling may alter cortical architecture during development. A key experiment to support this conclusion is testing whether neuronal morphology (e.g., neurite outgrowth) is altered in vivo in the cortex of these pups at postnatal developmental stages as compared to pups born to females who consumed regular diet.'

To dissect the effect of EPA-enriched diet on cortical architecture **we have performed an immunohistochemistry** using MAP2 and GAP43 on E18.5 mouse brains. As shown by our iTRAQ analysis, both MAP2 and GAP43 were found significantly down-regulated (**Fig. 6c**). Our histochemical analyses corroborated these findings, and showed changes in cortical organization (**Extended Data Fig. 6**). Thus, we are confident that the composition of the maternal diet during pregnancy is a key environmental variable to affect cortical development.

Q10: 'Minor points: 1. In the abstract (line 31) authors state .."guidance factors to promote neuronal differentiation".. is the intent here to say morphogenesis or neurite differentiation since neuronal cell differentiation occurs before gliogenesis?'

Thank you for this, **the sentence has been corrected.**

Q11: '2. In figures 1a-b scale bars, please provide numerical values (such as log2 cpm or fpkm) for the gene expression plots, as it is not possible to assess expression levels from min and max labels.'

These **figures have been replaced** to increase precision and clarity.

RE: Point-by-point replies to the Reviewers' comments;

Reviewer #1:

General: Thank you for reviewing our manuscript again and providing to-the-point questions to some of the details that remain lurking in a paper of this size. **We have carefully revised the manuscript to provide experimentally supported and valid answers to your queries, as follows. Please note that we did not spare man hours, energy, resources, and facilities to experimentally address your questions. Therefore, we are optimistic that you can approve our study for publication without further comments.**

Q1: 'Again, S100a6 expression in astrocytes is poorly presented in the revised manuscript. GFAP mRNA expression in Fig 2b is hardly visible in the cortical area, which is not surprising since cortical astrocytes express very low levels of GFAP. It is unclear why the authors did not use additional markers for astrocytes, such as *Aldh1l1*, *S100beta*, *APOE*, or *EAAT1/2*, to show colocalization of S100a6 with astrocytes. Moreover, the question of potential heterogeneous expression of S100a6 in astrocytic sub-populations was not properly investigated. This is an important point to support the authors' claim that S100a6 functions as an astrocytic signaling molecule in controlling neuronal protein turnover during general developmental stages.'

We have performed **additional neuroanatomy studies** to justify the co-expression of S100A6 with others marker you have requested: besides *Gfap*, *Aldh1l1*, *S100b*, and *ApoE* were hybridized. **Data were included in revised Figure 2 and Supplementary Figure 2.** Both E18.5 and P4 developmental time-points were systematically analyzed.

We appreciated your comment on the potential **heterogeneous expression of S100a6** in astrocyte subpopulations. Our present study was directed towards addressing intercellular communication by S100A6 under physiological condition. We think a detailed mapping study during development that charts the presence of S100A6 across brain areas remains beyond the scope of this mechanistic analysis. We also would caution that RNA-seq and anatomy might not match entirely given the 'reactive' or inducible nature and many state-changes of astrocytes during development (see our earlier papers: e.g., Romanov *et al.*, Nature 2020, Benevento *et al.*, Nature 2024). Therefore, we remain firm in our view that the major punchline shall remain the mechanistic analysis on how S100A6 could be used by astrocytes to modulate (or instruct) neuronal development, particularly neuritogenesis.

Figure 1 – S100A6 distribution in adult mouse brain, including corpus callosum-associated astrocytes.

Q2: 'S100a6 appears to be expressed in the oligodendrocyte lineage based on scRNA-seq data, but its expression pattern and contribution are not investigated or discussed. Without this information, how can the authors conclude that S100a6 is an astrocyte-derived factor?'

Thank you for this question. We think our interpretation that **S100A6 is an astrocyte-derived factor is entirely justified because**

it is found, and functionally mapped to astrocytes. We have never touched upon oligodendrocytes, and equally not claimed that S100A6 would be a general 'glial factor'. Thus, whether or not oligodendrocytes express S100A6 is unrelated to the conceptual thrust of this paper.

To satisfy your query experimentally, we would wish to add a few points: 1. RNA-seq (and RNA expression in general) can be misleading since there is no guarantee for protein expression even if mRNA could be detected. When looking at myelinated pathways in the mouse brain, we **do not find any classical depiction of S100A6 association to these** (see **Figure 1** herein). Likewise, we **did not observe any spatial arrangement for CaCyBp in neurons** that would suggest an instructive role for oligodendroglia in the brain (using myelin basic protein as a marker, *data not shown*). In sum, it may or may not be that oligodendrocytes indeed produce S100A6 and modulate, e.g., axonal integrity during some – as yet undefined – stage during the lifespan. Yet, we view this question as an entirely new study with new objectives that shall not be washed together with the present report. Therefore, **we have made a succinct point highlighting this possibility in the discussion.**

Q3: 'To claim that S100a6-HA localization in N2A cells is related to CaCyBp, the authors should knock down CaCyBp in N2A cells and report changes in S100a6-HA localization. Otherwise, the current EM and MS data cannot serve as evidence for the S100a6-CaCyBp interaction.'

Thank you for asking this. **Results are in Figure 5g.** Actually, the experiment return unexpected data. Your logic (and ours, too, initially) banked on CaCyBp acting as if areceptor. However, **CaCyBp is an internal binding partner. Therefore, its siRNA-mediated knock-down cannot (and did not) prevent S100A6 entry into the cells.** Instead, the lack of CaCyBp impaired S100A6 localization because its preferential association with the endoplasmic reticulum no longer existed. Thus, this experiment proved at least two points: CaCyBp being an intracellular binding partner without an effect on the plasmalemmal transfer of S100A6, and S100A6-CaCyBp interactions preferentially occurring in the vicinity of the endoplasmic reticulum. We did not address S100A6 transfer in either this or any other experiment because that is a whole new dimension of biochemical studies (transporters, binding sites) unrelated to our mechanistic analysis, and which deserves a detailed follow-up separately.

Q4: 'I cannot acknowledge the reduced neurite length shown in the images provided by the authors.'

We have **updated the figures on experiments focused on the modulation of neuritogenesis** after performing **a renewed round of immunocytochemistry.** In each figure panel, **we have clearly marked the neurite** that had been subjected to morphometric analysis (excluding the growth cone; e.g., **Figures 4 and 7**). The reason to exclude growth cones was that their morphological changes could allow several competing interpretations (most appealing would be that reduced or inhibited growth is associated with growth cone collapse; see our earlier work by Berghuis *et al.*, Science, 2007) yet would detract from the original hypothesis we addressed.

We have included **additional data demonstrating** that both silencing CaCyBp (shRNA) and exposing neurons to recombinant S100A6 led to reduced neuritogenesis, reinforcing our hypothesis that S100A6 is an inhibitor for neurite growth (**Figure 4g-j; Figure 7d-f**). We went as far as testing whether the combined effect of S100A6 exposure and CaCyBp gene silencing would be additive. Data were presented in revised **Figure 7.**

Q5: 'In Fig 3, the authors mentioned that CaCyBp-like immunoreactivity typically accumulated at subplasmalemmal positions in mitochondria. However, in Fig 4, they also mentioned that CaCyBp is mostly located in the endoplasmic reticulum. Which one is correct?'

We appreciate your attention to this critical detail. It was unfortunate if these data led to confusion. We have removed the data on S100A6 association to mitochondria, which is more sporadic, perhaps suggesting a cell-state-specific role. Since this was not addressed but the role of CaCyBp in modulating protein turnover, therefore **we have re-focused the revised manuscript entirely on CaCyBp localization near or to the endoplasmic reticulum (ER).** Data were included in revised **Figure 5.**

Q6: 'S100a6 treatment induces downregulation of CaCyBp, leading to a reduction in overall protein content. However, there is no strong data supporting that this general protein content reduction is actually due to UPR responses. Without functional interrogation of UPR responses, the S100a6-CaCyBp interaction and its role in protein homeostasis through UPR remain as a hypothesis.'

S100A6 treatment was extended with CaCyBp gene silencing. Both treatments evoked the same effect on slowed protein turnover. Moreover, we found reduced expression of GRP78, CHOP, and XBP1, as well as the downregulation of USP9x and USP7, the latter implicated in protein homeostasis through ER-associated degradation (ERAD). Thus, **our electron microscopy and molecular biology data define the subcellular site (organelle) and consequences of S100A6 action**. Together with increased protein lifetime (detected as increased content of posttranslational modifications), we bring together a number of findings that cumulatively support this hypothesis. We believe that the sum of these findings is stronger than, e.g., performing a non-physiological experiment with molecular sensors of protein ageing in some arbitrary cell line-based setting, that could be criticized for its limited relevance to natural processes during brain development.

Reviewer #2:

General: 'I congratulate the authors for submitted a much-improved manuscript. The data is now more robust and supports the majority of the conclusions. I thank the authors for addressing many of my comments.'

Thank you for your supportive and constructive comments on our submission. We have made every effort to justify our conclusions as per your comments made. **Please note that we did not spare man hours, energy, resources, and facilities to experimentally address your questions. Therefore, we are optimistic that you can approve our study for publication without further comments.**

Q1: 'My main concern was summarized in the following comment "More conclusive experiments are needed to demonstrate the astrocyte S100a6 – neuron CaCyBp exists and that there is a direct link between this and specific proteostatic mechanisms. At the moment all conclusions are based on correlative data. A combination of knock down/out experiments for the specific molecules involved will need to be performed in a cell specific manner to better understand the relationship between the molecular processes investigated in this study.". Although the authors have made a strong effort to improve the manuscript, I am afraid this manuscript is still lacking conclusive experiments that demonstrate a direct link between the astrocyte upregulation of S100a6 and neuronal downregulation of CaCyBp and other phenotypes. The manuscript is titled "Astrocytes modulate neuronal development by S100A6 signaling". Thus, it is essential that the data supports this idea.'

We have spared neither manpower/work hours nor resources to perform genetic experiments. data were included in **Figures 4-7 of the revised manuscript.**

Q2: 'Most of the in vivo experiments are done at E18.5, when there are few astrocytes and clear expression of S100a6 in other cell types (Fig. 1e-g).'

Indeed, astrocytogenesis is at early stages at E18.5 in mouse. This is why our study was systematically extended to postnatal day 4 (P4). Allow us to highlight though that single-cell RNA-seq clearly places astrocytes in the cerebral cortex at E18.5. These data are substantiated by our **additional palette of astrocyte markers (S100b, Aldh1l1, Apoe)** to localize S100a6 by *in situ* hybridization (revised **Figure 2**). These data significantly strengthened our conclusions.

Q3: 'The in situ hybridization figures (Fig. 2 and Extended Data 2) show that RNA of S100a6 and CaCyBp are in the same brain area, which is great, but they don't allow to assess in what cell type they are expressed.'

We have **significantly expanded the panel of markers** to allow for justified conclusions. For neurons, we used *Rbfox3*, which encodes NeuN, the prototypic neuronal marker (**Figure 2**). Moreover, we have included **quantitative data** on the distribution of CaCyBp in subsets of neurons.

Q4: 'The evidence in vitro is helpful. However, it is still lacking prove that the effects on neurons with astrocyte conditioned media comes from astrocyte-released S100a6. This is because, (1) glutamate and EPA are still in the media when collected so they could have a direct effect on the neurons, (2) many other molecules could be released by astrocytes in response to glutamate and EPA, (3) only by removing S100a6 from the media, before incubation or knocking down S100a6 in the astrocytes we would be able to conclude that S100a6 has an effect on neurons in vitro.'

This experiment has been done, a long and challenging effort. Data were shown in revised **Figure 7**, including many controls for both glutamate and eicosapentaenoic acid treatments. Thus, **the mechanistic basis of our hypothesis is firmly and conclusively supported.**

Q5: 'Thank you for doing the S100a4 experiment. It would be good to include this in the manuscript. Also, it would be useful to know if the neuron complexity is altered with S100a4. This will definitely conclude that there is a specific relationship between S100a6 and the morphology of neurons.'

Thank you for your appreciative feedback. Since S100A4 is not effective, **it does not affect neuronal morphology either**. We have made a **remark on this in the results section**. The figure was included as **Supplementary Figure 6** (also in the Extended Data File) for the manuscript.

Q6: 'An additional comment that emerged from the reviewed manuscript is that the authors say "EPA aids the formation of neuronal membranes, enhances membrane fluidity, and ensures the efficient transmission of neural signals. EPA also influences gene expression to facilitate the growth and differentiation of neurons, synaptogenesis, and myelination", and other similar statements. However, the experiments suggest EPA would enhance S100a6 release and as a consequence block neuronal branching. How are these two conclusions compatible? This should be discussed in the discussion.'

EPA is recognized for its beneficial effects on neuronal growth, and synaptogenesis. Its effects on membrane fluidity could well be dose dependent. At low concentrations, EPA promotes neurite outgrowth, consistent with its known neurotrophic properties. However, at higher concentrations, EPA could enhance the release of S100A6 from astrocytes, to inhibit neuronal morphogenesis. These two effects are not necessarily contradictory but rather reflect the pharmacological rationale how EPA influences neuronal development. We have **expanded the discussion to address this point**.

Q7: 'Overall, I think the results have potential interest, but they don't support the title "Astrocytes modulate neuronal development by S100A6 signaling", nor many of the conclusions that establish a direct link between astrocyte S100a6 and neuronal phenotypes. Either new experiments are included to conclude that, or a new title would be needed with rephrasing throughout the text to mention that the data suggests an interaction between astrocytes and neurons and that further experiments will be needed to establish robust conclusions.'

Modifying the title is certainly a possibility. **Nevertheless, we hope you agree with us that our revisions provide a sufficiently strong case for the original hypothesis to stand**, and, consequently, **for the original title**.

Reviewer #3:

General: 'The authors did a good job addressing some of the reviewers' comments by restructuring the manuscript text and figures and performing additional experiments. However, some points raised by this reviewer were not sufficiently addressed, raising concerns about the study's conclusions and interpretation. Points to be addressed are listed below.'

Thank you for your supportive and constructive comments on our submission. We have made every effort to address your remaining questions. **Please note that we did not spare man hours, energy, resources, and facilities to experimentally address your questions. Therefore, we are optimistic that you can approve our study for publication without further comments.**

Q1: 'In the rebuttal, the authors provide a good explanation for why S100A6 was chosen over other family members for further analysis (reviewer#3, Q2), however, the manuscript text states that it was chosen due to increased expression from E18 to P4 (lines 95-99). As previously mentioned, additional family members also show a sharp increase at P4. It would be helpful to edit the text to emphasize why this protein was selected for further analysis.'

We appreciate this request and **revised the manuscript accordingly**. While multiple S100 family members show an increase at P4, we specifically focused on S100A6 **due to its known binding partner (CaCyBp), and the purported downstream signaling system** (E3 ubiquitin ligases) that we used to infer function. In addition, **only S100A6 was modified by maternal diets during pregnancy, and particularly EPA**. The rationale was improved in the revised manuscript.

Q2 & Q3: 'The RNAseq data shows that cacybp is expressed by multiple cell types, including astrocytes at both E18 and P4 (Figure 1h, i). Yet the authors consider it to be specific to neurons. The in vivo validation experiments shown in Figure 2b are not convincing. Gfap signal is completely invisible or not shown and it is hard to see which cells co-express the markers. Figure 2c – bottom panel, does not appear like there is any colocalization based on the image. The findings in cell culture are encouraging, however, the validation in vivo needs to be further explored. The figure also lacks quantification (for example, how many map2 positive cells are colocalized with CaCyBp signal, or area or colocalized signal between 2 markers) and should be added.'

'The question whether S100A6 and CACYBP are expressed by astrocytes in vivo is insufficiently addressed. As mentioned in reviewer comments, gfap is not a good astrocyte marker for the cortex. The authors mention in the rebuttal that experiments using S100b and Aldh111 were performed in response to reviewer's comments – however, I could not find these in the revised manuscript. An experiment, similar to that shown in Figure 2c and extended data Figure 2c, d needs to be performed with S100b or Aldh111 as an astrocytic marker (similarly to what was done using neuron as neuronal marker). The experiments in extended data Figure 2g are showing that these markers are expressed by astrocyte in culture, however, it does not prove the astrocytic origin of S100A6 or CACYBP in vivo.'

We have indeed performed additional **in situ hybridization experiments** systematically at E18.5 and P4, with all data included in revised **Figure 2**. Besides *Gfap*, *S100b*, *ApoE*, and *Aldh111* were used. These data further support that S100A6 is expressed by astrocytes in contrast to CaCyBp, which is primarily found in neurons.

We have quantified the number of **MAP2-positive and CR-positive neurons** that contained Ca-CyBP to support our hypothesis (**Figure 2c**).

Text edits ensure that our findings are clearly presented and integrated into the manuscript.

Q4: 'Extended data Figure 2e – the zoomed-in region from box labeled 1 is not showing the same area of the image.'

The figure has been corrected, and fields of view matched.

Q5: 'For all images in Figure 2 and extended data Figure 2 – it would be helpful to add arrows to the images to guide the reader to the relevant cell type / signal that is discussed in the text or figure legend.'

We have updated the figures.

Q6: 'The images of neurons provided in Figure 3h (in response to reviewer's comments Q7) are only shown for an experiment in which *cacybp* is downregulated. The images shown (Figure 3h) are hard to interpret as only the length of branches is analyzed. However, it looks from the image that the neuron treated with *cacybp* shRNA shows increased branch number. As the implications of S100A6-CACYBP signaling on neuronal morphogenesis is one of the key findings of this study, performing a more in-depth analysis and providing stronger evidence for morphological changes upon these treatments is very important. Authors should provide images of neurons treated with S100A6 (Figure 3f) so that they can be compared with the morphological differences shown upon the knockdown of *cacybp*. Sholl analysis to determine neuronal morphological complexity would be beneficial for these experiments.'

We performed additional analyses using both **qPCR and immunocytochemistry** to even better characterize the effects of S100A6 treatment and/or CaCyBp knock-down on neuronal morphology. Our **qPCR data** show that treatment with both **S100A6** and shRNA for *Cacybp* either individually or in combination, **decreased the expression of cytoskeletal genes**, including *Tubb3* and *Actb*.

Immunocytochemistry showed that both **shRNA-mediated *Cacybp* knock-down and S100A6 treatment (alone or in combination) resulted in both significantly shorter neurites and reduced branch formation (Figure 4).**

As requested, we have included images of **neurons treated with S100A6 (Figure 4g)** to allow for a direct comparison with the morphological changes observed upon *Cacybp* knock-down. These additional experiments comprehensively characterize the effects of S100A6.

Q7: 'The downregulation in *map2* and *gap43* signals shown in extended data Figure 6 is quite dramatic. Do these mice exhibit severe brain developmental abnormalities after birth?'

To address this question, we have included data examining both **cell distribution** (by EdU pulsing at E14.5 and sacrifice at E18.5) during brain development and **behavioural outcomes** in adulthood (**Figure 8a,g**). Our results indicate that there is an accumulation of neurons in deep layers, suggesting altered cortical lamination during antenatal development.

We have also included **behavioural data from P75 animals**, which were reared on control chow after weaning. We observed an **anxiety-like phenotype in the elevated plus maze (Figure 8g)**. These findings suggest long-lasting consequences of dietary manipulations during pregnancy.

Q8: 'The authors state that the number and positioning of the neurons is altered upon the experimental diet, based on downregulated expression of *map2* and *gap43* (lines 307-308). It is unclear how this conclusion was reached based on the data, since the number of *dapi+* cells is not altered. A detailed analysis of neuronal numbers and positions within the cortex should be conducted, or the text should be revised to more accurately describe the experimental finding.'

In our experiments, all dams were pulsed with **EdU** to allow us to test cell distribution in the cerebral cortex. **These data were added to Figure 8a**, to quantitatively show that the experimental diet during pregnancy affected cortical architecture.

RE: Point-by-point replies to the Reviewers' comments;

Reviewer #1:

Thank you for appreciating the extent of our revisions. We were glad to learn your remaining comments, which indeed will help to pitch the manuscript appropriately without unnecessarily exaggerating our findings. Please find our specific responses to your queries as follows:

Q1: 'The authors emphasize that S100A6 could serve as an astrocyte-specific gliotransmitter. However, based on their scRNA-seq and fluorescent in situ hybridization data, it is evident that S100A6 is not exclusively expressed in astrocytes, but also in other cell types. Given that S100A6 can be secreted by various cells during both embryonic and postnatal development, how can the authors claim that S100A6 functions as an astrocyte-specific gliotransmitter?'

We think this is the classical discrepancy between single-cell RNA-seq and neuroanatomy: mRNA is not necessarily translated into protein. Nevertheless, we agree with your implicit notion that this work does not aim to describe the brain-wide mapping of S100A6 expression. Therefore, **we have refrained from qualifying S100A6 as a gliotransmitter.**

Q2: 'In addition, the conclusions drawn from the proteostasis-related experiments and eicosapentaenoic acid (EPA) treatment appear to be largely correlative, lacking direct causal evidence. Most of the proteins examined seem to be downregulated following S100A6 treatment or Cacybp knockdown. How, then, can the authors attribute these effects specifically to the unfolded protein response (UPR) or ubiquitin-proteasome system (UPS)?'

At no point have we concluded that EPA exclusively affects the UPR/UPS systems of developing neurons. Because S100A6-CaCyBp were the only complementary ligand-binding partner pair that changed upon EPA (whilst acknowledging potential shortcomings of iTRAQ proteomics), and neuronal migration seemed to slow in EPA-exposed fetuses, we felt the conceptual framework was correct to implicate S100A6-mediated changes in neuronal differentiation for the phenotype be appropriately explained. To comply with your request, we have removed any statement on causation.

Q3: 'Finally, although the levels of S100A6 and Cacybp were altered by EPA treatment, this alone does not support the conclusion that the observed outcomes—particularly in the in vivo experiments—are mediated through the S100A6–Cacybp signaling pathway.'

Thank you for remarking on this point. As stated above, we have removed any point of causation from the manuscript.

Reviewer #2:

Feedback: 'I thank the authors for carefully considering my comments and the effort they have put in addressing them. I think the manuscript has greatly improved and I would be happy for it to be published. Thanks!'

Thank you for approving our submission for publication.

Reviewer #3:

Feedback: 'The authors have adequately addressed all my comments. I support the acceptance of this manuscript for publication.'

Thank you for supporting the publication of our work.